# MaD-Scientist: AI-based Scientist solving Convection-Diffusion-Reaction Equations Using Massive PINN-Based Prior Data

## Abstract

Large language models (LLMs), like ChatGPT, have shown that even trained with noisy prior data, they can generalize effectively to new tasks through in-context learning (ICL) and pre-training techniques. Motivated by this, we explore whether a similar approach can be applied to scientific foundation models (SFMs) for solving PDEs. Our methodology is structured as follows: (i) we collect low-cost physics-informed neural network (PINN)-based approximated prior data in the form of solutions to partial differential equations (PDEs) constructed through an arbitrary linear combination of mathematical dictionaries; (ii) we utilize Transformer architectures with self and cross-attention mechanisms to predict PDE solutions without knowledge of the governing equations in a zero-shot setting; (iii) we provide experimental evidence on the one-dimensional convection-diffusion-reaction equation, which demonstrate that pre-training remains robust even with approximated prior data, with only marginal impacts on test accuracy. Notably, this finding opens the path to pre-training SFMs with realistic, low-cost data instead of (or in conjunction with) numerical high-cost data. These results support the conjecture that SFMs can improve in a manner similar to LLMs where fully cleaning the vast set of sentences crawled from the Internet is nearly impossible.

## 1 Introduction

In developing large-scale models, one fundamental challenge is the inherent noisiness of the data used for training. Whether dealing with natural language, scientific data, or other domains, large datasets almost inevitably contain noise. Large language models (LLMs), such as ChatGPT, present an interesting paradox: despite being trained on noisy datasets, they consistently produce remarkably clean and coherent output. This observation raises an important question for the development of scientific foundation models (SFMs): Can an SFM, like an LLM, learn from noisy data and still generate accurate, dynamic results for solving PDEs, one essential task of sciences?

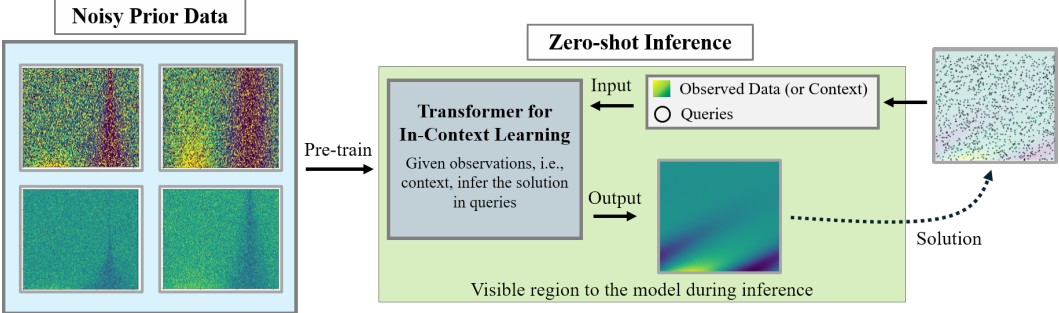

Figure 1: **An end-to-end schematic diagram of our model.** Our model performs in-context learning based on the given observations (i.e., context) to infer the solution. Even when trained with noisy PINN-prior, our model can obtain clean solutions due to its Bayesian inference capability.

In recent years, LLMs have revolutionized the field of natural language processing by introducing highly flexible and scalable architectures (Brown et al., 2020; Kaplan et al., 2020; Touvron et al., 2023; Frieder et al., 2023; Chowdhery et al., 2023). Notably, the in-context learning (ICL) paradigm has demonstrated powerful generalization capabilities, enabling LLMs to adapt to new tasks without explicit fine-tuning (Brown et al., 2020; Radford et al., 2019; Dai et al., 2023; Gruver et al., 2023). This success has motivated the application of such foundation models across a variety of domains (Xu et al., 2024; Xie et al., 2024; Yang et al., 2023a). Scientific machine learning (SciML) is one such emerging domain which merges physics-based models with machine learning methodologies (Raissi et al., 2019; Willard et al., 2022; Subramanian et al., 2023; Kim et al., 2024; 2023; Choi et al., 2024). SciML aims to leverage the power of machine learning to solve complex scientific problems, including those governed by partial differential equations (PDEs). Recent efforts in this direction have led to the development of foundation models specifically designed for scientific tasks, called SFMs (Yang et al., 2023b; Xie et al., 2024; Yang et al., 2023a; Moor et al., 2023; Bodnar et al., 2024; Herde et al., 2024). These models aim to generalize across a wide range of scientific problems using prior data, much like how LLMs generalize across various language tasks. For example, the versatility of in-context operator networks (ICONs), as illustrated in studies like Yang & Osher (2024) and Yang et al. (2023b), underscores their generalization capabilities in various PDE-related tasks, particularly in the context of few-shot learning. Moreover, the integration of in-context operator learning into multi-modal frameworks, as demonstrated by ICON-LM (Yang et al., 2023c), has pushed the boundaries of traditional models by combining natural language with mathematical equations. Additionally, several other studies have focused on solving a family of PDEs with a single trained model (Cho et al., 2024). However, all these studies are limited in their ability to fully harness the capabilities of large foundation models. Our methodology, called **Ma**ssive prior **D**ata-assisted AI-based **Scientist** (MaD-Scientist), addresses these limitations and offers significant advantages in the following four aspects.

**No prior knowledge of physical laws**  Our goal is to predict solutions from observed quantities, such as velocity and pressure, without relying on governing equations, a common challenge in many real-world scenarios (Lee & Cant, 2024; Nicolaou et al., 2023; Rouf et al., 2021; Beck & Kurz, 2021; Chien et al., 2012). In complex systems, such as those governing semiconductor manufacturing, the exact governing equations are often unknown and may change over time (Chien et al., 2012; Quirk & Serda, 2001). Therefore, excluding these equations from the model input is a strategic choice aimed at enhancing the applicability of our method across various domains.

**Zero-shot inference**  Our goal is to achieve zero-shot inference for predicting PDE solutions. For instance, ICON-LM requires few-shot "demos"[1] for an unknown target operator before making predictions. In contrast, our foundation model eliminates the need for such demos, as collecting them implies that inference cannot occur until these few-shot examples are available; see e.g., Figure 1. Our approach is designed to enable immediate inference as soon as the model is queried.

**Bayesian inference**  We incorporate Bayesian inference into the prediction process by leveraging prior knowledge obtained from numerical solutions in PDE dictionaries. This approach allows the model to make more accurate and well-informed predictions by defining a prior distribution over unseen PDE coefficients. During training, the model learns to capture relationships among known data points using self-attention mechanisms, while cross-attention enables it to extrapolate and infer solutions for new, unseen points. When tested, the model utilizes this prior data to generalize effectively to novel data points, achieving zero-shot predictions without the need for additional fine-tuning.

**Approximated prior data**  For LLMs, one of the most challenging steps is collecting prior data, which typically involves crawling and cleaning sentences from the Internet. However, this process is far from perfect due to two key issues: (i) the Internet, as a data source, is inherently unreliable; (ii) cleaning such vast amounts of data requires significant manual effort. Consequently, LLMs are often trained on incomplete or imperfect prior data. Remarkably, this realistic yet critical issue has been largely overlooked in the literature on SFMs, despite their similarities to LLMs. For example, when generating data using numerical solvers for PDEs without known analytical solutions, numerical errors inevitably arise, manifesting as a form of measurement noise — for this, we conducted pre-

---

[1]In ICON and ICON-LM, a demo means a set of (input, output) pairs of an operator to infer.

liminary experiments for training SFMs with noisy data in Appendix G, which shows the possibility of training SFMs with data inexact to some degree.

Moreover, numerical solvers running on large-scale servers are frequently expensive and time-consuming and they are typically optimized towards certain types of PDEs, e.g., the finite-difference time-domain (FDTD) method for Maxwell's equations. In this work, we are the first to explore the potential of pre-training SFMs with PINN-based low-cost/noisy/approximated data.

**CDR Study**   For our empirical studies, we use a family of the convection-diffusion-reaction (CDR) equation with various types of reaction, which serves as a paradigm problem representing generic elliptic equations. By solving the CDR equation, our approach can be extended to a wide range of other problems. We compare our method with two state-of-the-art machine learning techniques for solving parameterized PDEs. Additionally, we introduce three different types of noise into the numerical solutions of the CDR equation. Our approach not only outperforms the two baseline methods but also demonstrates stable performance, even when noise is added to the prior data during pre-training.

## 2 BACKGROUND

Consider a sequence of pairs $(X_1, Y_1), (X_2, Y_2), \ldots$, each within the measurable space $(\mathcal{X} \times \mathcal{Y}, \mathcal{A})$, where $X_i$ represents the spatiotemporal coordinate, $Y_i$ denotes the corresponding solution in this paper's context and $\mathcal{A}$ denotes the Borel $\sigma$-algebra on the measurable space $\mathcal{X} \times \mathcal{Y}$. For simplicity, we adopt this notation in this section. These pairs are drawn from a family of probability density distributions $\{p_q : q \in \mathcal{Q}\}$, commonly referred to as the *statistical model*, where $\mathcal{Q}$ represents the *parameter space* equipped with a $\sigma$-algebra $\mathcal{B}$ ensuring that the mappings $q \mapsto p_q(x, y)$ are measurable. The true underlying density function $\pi$ is a member of $\mathcal{Q}$, and the pairs $(X_i, Y_i)$ are sampled according to $p_\pi$. Lacking information about $\pi$, we adopt a Bayesian framework to establish a prior distribution $\Pi$ which is defined as probability measure on $(\mathcal{Q}, \mathcal{B})$. Then we have, for any measurable set $A \in \mathcal{B}$,

$$\Pi(A \mid X, Y) = \frac{\int_A p_q(X, Y) \mathrm{d}\Pi(q)}{\int_\mathcal{Q} p_q(X, Y) \mathrm{d}\Pi(q)}. \tag{1}$$

Let us adopt the notation $p_q = q$. This prior is updated with the observed data to form the posterior distribution, which is defined as

$$\Pi(A \mid D_n) = \frac{\int_A L_n(q) \mathrm{d}\Pi(q)}{\int_\mathcal{Q} L_n(q) \mathrm{d}\Pi(q)}, \tag{2}$$

where $L_n(q) = \prod_{i=1}^n \frac{q(X_i, Y_i)}{\pi(X_i, Y_i)}$ for $A \subset \mathcal{Q}$ and $D_n = \{(X_i, Y_i)\}_{i=1}^n$. The resulting posterior density is

$$q_n(X, Y \mid D_n) = \int_\mathcal{Q} q(X, Y) \mathrm{d}\Pi(q \mid D_n), \tag{3}$$

and the posterior predictive distribution (PPD) is formulated as

$$\pi(y \mid x, D_n) = \int_\mathcal{Q} q(y \mid x) \, \mathrm{d}\Pi(q \mid D_n). \tag{4}$$

The behavior of $D_n$ plays a crucial role in this formulation. As noted by Walker (2004b;a); Blasi & Walker (2013); Walker (2003); Nagler (2023), for a well-behaved prior, the PPD converges toward $\pi$ as $n$ increases. This aligns with findings in Blasi & Walker (2013), demonstrating that in well-specified scenarios, strong consistency is achieved as

$$\Pi^n \left( \{q : H(\pi, q) > \epsilon\} \right) \to 0 \quad \text{almost surely}, \tag{5}$$

for any $\epsilon > 0$, where $\Pi^n(A) = \int_A d\Pi(q \mid D_n)$ is the posterior measure and $H$ is the Hellinger distance defined by

$$H(p, q) = \left( \int_{\mathcal{X} \times \mathcal{Y}} (\sqrt{p} - \sqrt{q})^2 \right)^{1/2}.$$

**Theorem 2.1.** *Suppose that for any $\epsilon > 0$, there exists a Transformer parameterized by $\hat{\theta}$ such that*

$$\mathbb{E}_x \left[ KL \left( p_{\hat{\theta}}(\cdot \mid x, D_n), \pi(\cdot \mid x, D_n) \right) \right] < \epsilon,$$

*for any realization of $D_n$. If the posterior consistency condition equation 5 holds, and for any $q \in \mathcal{Q}$, $q(x) = \pi(x)$ almost everywhere on $\mathcal{X}$, then the following holds almost surely (see Appendix A for proof):*

$$\mathbb{E}_x \left[ H \left( p_{\hat{\theta}}(\cdot \mid x, D_n), \pi(\cdot \mid x) \right) \right] \xrightarrow{n \to \infty} 0.$$

This result demonstrates sensitivity of the neural network's posterior distribution approximation to the data size $D_n$. As the data size increases, the network becomes increasingly sensitive to the posterior distribution, converging to the expected value under the prior distribution. This sensitivity to the data reflects the consistency and robustness of the Bayesian inference process.

Our model leverages this observation by performing Bayesian inference that incorporates prior data, allowing it to infer spatiotemporal points that align with an appropriate partial differential equation (PDE) solution under given spatiotemporal conditions. Experimental results in (Appendix M) confirm this behavior, showing how, as $D_n$ increases, the network's solution converges to the true underlying solution.

## 3 METHODS

Suppose the dataset $D = \{(X_i, T_i, Y_i)\}_{i=1}^n$ is independently and identically distributed (i.i.d.) and sampled from a distribution $q_\alpha$, where $\alpha$ is the parameter vector representing the coefficients governing the PDE dynamics, including convection, diffusion, and reaction terms. Specifically, $Y_i \sim u(X_i, T_i \mid \alpha) + \text{noise}$, where the noise represents the difference between the PINN-predicted solution $\tilde{u}(\alpha)$ and the true solution $u(\alpha)$. The PPD of the solutions given the dataset can be expressed as

$$q(y \mid x, t, D) = \int_H q_\alpha(y \mid x, t) \, d\Pi(q_\alpha \mid D), \tag{6}$$

which represents the likelihood distribution of $y$ given $D$, capturing the most probable solution distribution for the given parameter $\alpha$. In this work, we aim to predict the solution from $D$ by minimizing the mean squared error (MSE) between the PPD-derived solution and the true solution, even in the presence of noise. This requires constructing a prior over the PDE solution space, which is detailed next.

**Benchmark PDE**   The following one-dimensional convection-diffusion-reaction (CDR) equation is used for the benchmark PDE,

$$\text{1D CDR: } u_t + \beta u_x - \nu u_{xx} - \rho f(u) = 0, \quad x \in [0, 2\pi], t \in [0, 1], \tag{7}$$

where $f : \mathbb{R} \to \mathbb{R}$ is a reaction term such as Fisher, Allen-Cahn and Zeldovich. This equation consists of three key terms with distinct properties: convective, diffusive, and reactive, making it an ideal benchmark problem. It is commonly used in the PINN literature due to the diverse dynamics introduced by its three parameters: $\beta, \nu,$ and $\rho$, which include various failure modes (Krishnapriyan et al., 2021). To our knowledge, however, our work is the first predicting all those different reaction terms with a single model.

In this paper, we use the following dictionary of CDR-related terms, incorporating a linear combination of $J$ nonlinear reaction terms, for generating prior data.

$$u_t = \mathcal{N}(\cdot), \quad \mathcal{N}(t, x, u, \beta, \nu, \rho_1, \cdots, \rho_J) = -\beta u_x + \nu u_{xx} + \sum_{j=1}^{J} \rho_j f_j(u), \tag{8}$$

where each $f_j$ represents specific reaction term. This expansion allows for the introduction of diverse reaction dynamics. One can solve CDR equations with numerical solvers. In this work, however, we are interested in building low-cost PINN-based prior data. In the future, one may need to build prior data for not only CDR but also many other equations for which none of analytical/numerical solutions are obtainable in a low-cost manner, e.g., Naiver-Stokes equations. We think our PINN-based prior data will play a crucial role in such a case.

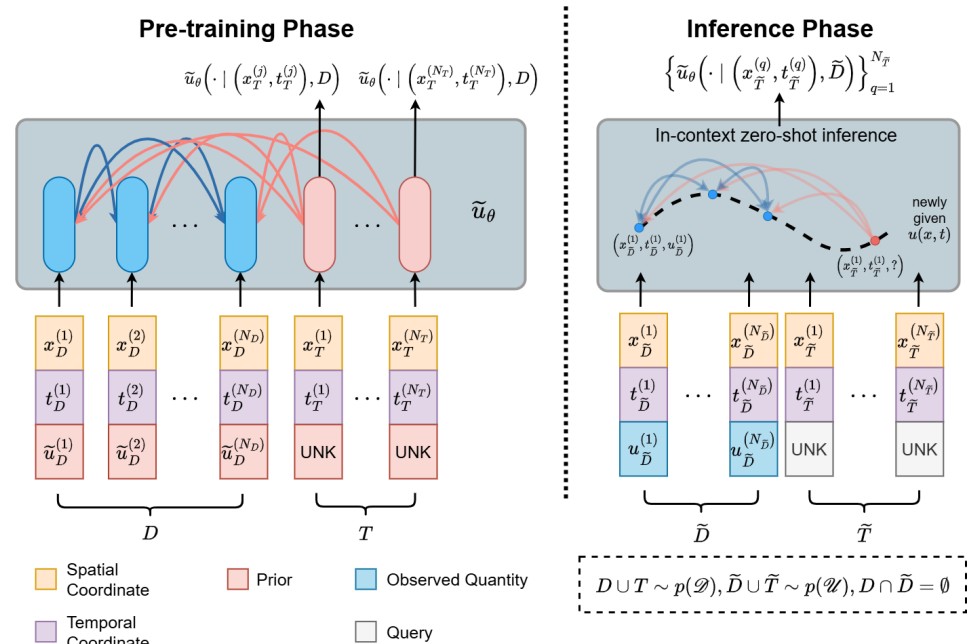

Figure 2: **A schematic diagram of Transformer.** *(Left)* The Transformer $\tilde{u}_\theta$ takes prior of solution-known $D$ and querying task $T$ drawn from the prior distribution $\mathcal{D}$ and infers solutions of the queried points in the training phase. ICL is leveraged with a self-attention among $D$ (blue rods) and a cross-attention from $T$ (red rods) to $D$. *(Right)* In the testing phase, $\tilde{u}_\theta$ takes an input of unseen data $\widetilde{D}$ and $\widetilde{T}$ drawn from the ground truth distribution $\mathcal{U}$, and the model predicts the queried points $\widetilde{T}$.

**PINN-Prior of PDE Solution Space**     To approximate the solution space for PDEs, we construct a parameter space, $\Omega$, which is the collection of coefficients in equation 8:

$$\Omega = \{\alpha := (\beta, \nu, \rho_1, \cdots, \rho_J)\}, \tag{9}$$

which has a dictionary form. Consequently, the target exact prior $\mathcal{U}$ represents the collection of solutions $u(\alpha)$ at equation 8 for each parameter $\alpha \in \Omega$, where $\mathcal{X}$ and $\mathcal{T}$ correspond to the spatial and temporal domains of interest, respectively

$$\mathcal{U} = \bigcup_{\alpha \in \Omega} \{u(\alpha) \,|\, u_t = \mathcal{N}(t, x, u, \alpha)\}, \quad \mathcal{U} : \mathcal{X} \times \mathcal{T} \to \mathbb{R}. \tag{10}$$

Since the target exact prior data $\mathcal{U}$ is hard to obtain, we instead use a PINN-prior $\mathcal{D}$ that closely approximates $\mathcal{U}$ as follows. Suppose $\tilde{u}(\alpha)$ is the prediction by PINN (Appendix E) which is trained to predict the PDE $u_t = \mathcal{N}(\cdot)$. The PINN-prior $\mathcal{D}$ is a collection of approximated solutions $\tilde{u}(\alpha)$ for each $\alpha \in \Omega$,

$$\mathcal{D} = \bigcup_{\alpha \in \Omega} \{\tilde{u}(\alpha)\}, \quad p(\mathcal{D}) \sim p(\mathcal{U}). \tag{11}$$

Subsequently, the model learns the PPD of the generated prior $p(\mathcal{D})$ through ICL.

**Training**     From a given parameter space $\Omega$, the parameter $\alpha$ is randomly drawn i.i.d. from $\Omega$. This method is adopted from meta learning (Finn et al., 2017) which optimizes the model parameter to adapt to various tasks, in our case the prediction over wide prior space $\mathcal{D}$ expressed as a dictionary over $\alpha$. After that, the previous $\tilde{u}(\alpha)$ is then given as an input to Transformer $\tilde{u}_\theta$ to minimize the mean square error (MSE) at the predicted points, see equation 12. The MSE loss criterion is proposed as the Transformer's task is to perform regression of the solution over the spatial and temporal domain for given $\tilde{u}(\alpha)$,

$$L_\alpha = \frac{1}{N_T} \sum_{j=1}^{N_T} \left[ \tilde{u}_\theta(x_T^{(j)}, t_T^{(j)} \mid D) - \tilde{u}(x_T^{(j)}, t_T^{(j)}) \right]^2. \tag{12}$$

Table 1: Major comparisons between Hyper-LR-PINN, P$^2$INN, and our model. While both Hyper-LR-PINN and P$^2$INN require the knowledge of governing equation, our model only needs observed quantities. The notations used in the table are fully aligned with those in Figure 2.

| Properties | Hyper-LR-PINN | P$^2$INN | Ours |
|---|---|---|---|
| Target function | $u(x, t; \alpha \in \Omega)$ | $u(x, t; \alpha \in \Omega)$ | $u(x, t)\|_{\mathcal{D}}$ |
| Governing equation $\mathcal{N}(\cdot)$ given | ✓ | ✓ | ✗ |
| Train dataset | $D \cup T \cup \widetilde{D}$ | $D \cup T \cup \widetilde{D}$ | $D \cup T$ |
| Test dataset | $\widetilde{T}$ | $\widetilde{T}$ | $\widetilde{D} \cup \widetilde{T}$ |
| Dataset with a solution | None | None | $D, T, \widetilde{D}$ |

**Evaluation**   To illustrate the model's zero-shot learning capability in scenarios commonly encountered in practical applications, we assess the model's performance using data $\tilde{D}$ sampled i.i.d. from $\mathcal{U}$, not overlapping with the training set $D \cup T$. For evaluation, we employ both $L_1$ absolute and relative $L_2$ errors between the model's predicted solutions for test queries and the numerically computed ground truth. Errors are then averaged over the target parameter space $\Omega$ used during training.

## 4   EXPERIMENTS

Our experiment section is divided into two phases: in the first phase, we conduct a focused study with the basic reaction term, Fisher, to understand the base characteristics of SFMs, and in the second phase we conduct comprehensive studies with various reaction terms.

### 4.1   EXPERIMENTAL SETUP

**Baseline methods**   We compare our model with 2 baselines: Hyper-LR-PINN (Cho et al., 2023) and P$^2$INN without fine tuning (Cho et al., 2024). Both models are parametrized PINNs designed to learn parameterized PDEs. Hyper-LR-PINN emphasizes a low-rank architecture with a parameter hypernetwork, while P$^2$INN focuses on a parameter-encoding scheme based on the latent space of the parameterized PDEs.

Following this, as shown in Figure 2, the model takes $D \cup T \sim p(\mathcal{D})$ in training phase and $\widetilde{D} \cup \widetilde{T} \sim p(\mathcal{U})$ in testing phase. In addition, the dataset $D \cup T$ requires the prior $\widetilde{u}$, and $\widetilde{D}$ requires the solution $u$. For a fair comparison, we use $D$, $T$, and $\widetilde{D}$ as the training dataset for both Hyper-LR-PINN and P$^2$INN. Notably, while Hyper-LR-PINN and P$^2$INN do not rely on solution points during training and testing, our model operates without any knowledge of the governing equation $\mathcal{N}(\cdot)$. This setup ensures a valid and balanced comparison (Table 1). The additional comparison details are elaborated in Appendix C, providing further insights into the distinctions between these models.

**Training algorithm**   The concrete flow of training phase is described in Algorithm at Appendix F.

### 4.2   FOCUSED STUDY TO BETTER UNDERSTAND SFMS' BASE CHARACTERISTICS

In this section, we employ six different dynamics derived from the 1D CDR equation with a Fisher reaction term, $u_t + \beta u_x - \nu u_{xx} - \rho u(1 - u) = 0$ (Appendix **??**). We begin with an in-depth study using the Fisher reaction term, chosen for its simplicity among the reaction terms, which has been extensively studied in population dynamics (Al-Khaled, 2001). This allows us to better understand the core characteristics of the SFM, facilitating a more effective analysis of the model's behavior.

#### 4.2.1   TIME DOMAIN INTERPOLATION FOR SEEN PDE PARAMETERS WITH A NUMERICAL PRIOR

We first verify the ICL capability of Transformer with a numerical prior, i.e., $\tilde{u}(\alpha)$ equals to the solution $u$ of the PDE $u_t = \mathcal{N}(t, u, x, \alpha)$, before we dive into using a PINN prior. For each equation, we set the parameter space $\Omega$ with three different coefficient $(\beta, \nu, \rho)$ range: $([1, 5] \cap \mathbb{Z})^m$, $([1, 10] \cap \mathbb{Z})^m$, and $([1, 20] \cap \mathbb{Z})^m$, where $m$ is the number of nonzero coefficients.

Table 2: The $L_1$ absolute and $L_2$ relative errors over the 1D-CDR equation using a numerical prior. P$^2$INN is tested without fine-tuning, and *-marked cases are evaluated with a reduced number of parameters due to the extensive computational requirements.

| System | Coefficient range | Hyper-LR-PINN | | P$^2$INN | | Ours | |
|---|---|---|---|---|---|---|---|
| | | Abs.err | Rel.err | Abs.err | Rel.err | Abs.err | Rel.err |
| **Convection** | $\beta \in [1,5] \cap \mathbb{Z}$ | **0.0104** | **0.0119** | 0.0741 | 0.1020 | 0.0192 | 0.0184 |
| | $\beta \in [1,10] \cap \mathbb{Z}$ | **0.0172** | **0.0189** | 0.1636 | 0.1801 | 0.0250 | 0.0251 |
| | $\beta \in [1,20] \cap \mathbb{Z}$ | **0.0340** | **0.0368** | 0.2742 | 0.2743 | 0.0764 | 0.0864 |
| **Diffusion** | $\nu \in [1,5] \cap \mathbb{Z}$ | 0.0429 | 0.0570 | 0.3201 | 0.3652 | **0.0096** | **0.0120** |
| | $\nu \in [1,10] \cap \mathbb{Z}$ | 0.0220 | 0.0282 | 0.3550 | 0.4029 | **0.0108** | **0.0137** |
| | $\nu \in [1,20] \cap \mathbb{Z}$ | 0.1722 | 0.1991 | 0.4553 | 0.5166 | **0.0095** | **0.0134** |
| **Reaction** | $\rho \in [1,5] \cap \mathbb{Z}$ | 0.0124 | 0.0428 | 0.0109 | 0.0354 | **0.0102** | **0.0154** |
| | $\rho \in [1,10] \cap \mathbb{Z}$ | 0.2955 | 0.3562 | 0.0192 | 0.0708 | **0.0129** | **0.0202** |
| | $\rho \in [1,20] \cap \mathbb{Z}$ | 0.7111 | 0.7650 | 0.1490 | 0.2915 | **0.0160** | **0.0322** |
| **Convection-Diffusion** | $\beta,\nu \in [1,5] \cap \mathbb{Z}$ | **0.0046** | **0.0055** | 0.1329 | 0.1554 | 0.0195 | 0.0231 |
| | $\beta,\nu \in [1,10] \cap \mathbb{Z}$ | 0.0268 | 0.0295 | 0.1609 | 0.1815 | **0.0211** | **0.0274** |
| | $\beta,\nu \in [1,20] \cap \mathbb{Z}$ | *0.1487 | *0.1629 | 0.1892 | 0.2044 | **0.0226** | **0.0305** |
| **Reaction-Diffusion** | $\nu,\rho \in [1,5] \cap \mathbb{Z}$ | 0.0817 | 0.1160 | 0.0579 | 0.1346 | **0.0139** | **0.0189** |
| | $\nu,\rho \in [1,10] \cap \mathbb{Z}$ | 0.0317 | 0.0446 | 0.4398 | 0.5457 | **0.0122** | **0.0189** |
| | $\nu,\rho \in [1,20] \cap \mathbb{Z}$ | *0.3228 | *0.3844 | 0.1513 | 0.2955 | **0.0165** | **0.0331** |
| **Convection-Diffusion-Reaction** | $\beta,\nu,\rho \in [1,5] \cap \mathbb{Z}$ | 0.0231 | 0.0307 | 0.0418 | 0.0595 | **0.0143** | **0.0209** |
| | $\beta,\nu,\rho \in [1,10] \cap \mathbb{Z}$ | *0.3135 | *0.3732 | 0.0367 | 0.0624 | **0.0276** | **0.0411** |
| | $\beta,\nu,\rho \in [1,20] \cap \mathbb{Z}$ | *0.9775 | *0.9958 | 0.0446 | 0.1211 | **0.0159** | **0.0310** |
| **Statistics** | Average | 0.1805 | 0.2033 | 0.1709 | 0.2222 | **0.0196** | **0.0267** |
| | Standard Deviation | 0.2581 | 0.2727 | 0.1423 | 0.1549 | **0.0147** | **0.0164** |

The Transformer $\widetilde{u}_\theta$ is trained with $D \cup T \subseteq u(\alpha)$ where $\alpha \in \Omega$ is selected uniformly at random for each epoch. After that, we test $\widetilde{u}_\theta$ with $\widetilde{D} \cup \widetilde{T} \subseteq u(\alpha)$ for all $\alpha \in \Omega$ and evaluate average $L_1$ absolute and $L_2$ relative errors (Table 2).

We highlight two key observations: First, our model outperforms baseline models applied to diffusion, reaction, reaction-diffusion, and convection-diffusion-reaction systems. Second, it demonstrates stable performance across a wide range of coefficient values. For instance, all baselines show difficulties in predicting accurate solutions for high coefficients, especially in diffusion and reaction systems, while ours do not. When we measure the standard deviation of $L_2$ relative error over three coefficient range for diffusion system, ours have $9.1 \times 10^{-4}$ while others show $10^{-2}$ scale value. These observations not only verify the effectiveness of the Transformer's ICL capability, but also suggest its potential to handle larger parameter space $\Omega$.

### 4.2.2 TIME DOMAIN INTERPOLATION FOR SEEN PDE PARAMETERS WITH A PINN-PRIORS

The Transformer has demonstrated strong ICL capabilities when trained with numerical priors. Our main focus now is to determine if this same success can be achieved using a PINN-prior. As outlined in Appendix C, our preliminary results show that the Transformer remains robust even when numerical priors are subject to various types of noise. Building on this, we examine how the model performs when mixing low-cost PINN-priors with numerical priors in different proportions, assessing its stability and robustness when incorporating PINN-priors.

Specifically, we train the model using the convection, diffusion, and Fisher reaction equations with integer coefficients ranging from 1 to 20. For each equation, we evaluate the model with a prior that is a mixture of PINN-prior and numerical prior in varying ratios: 0%, 20%, 40%, 60%, 80%, and 100% PINN-priors. Table 3 indicates the $L_1$ absolute and $L_2$ relative errors for each setup compared to the baseline results in Section 4.2.1. Furthermore, the average error of the PINN-prior, compared to the numerical solution, is presented to demonstrate the quality of the PINN-prior.

As a result, mixing PINN-priors with numerical priors does not significantly impact performance, as the $L_1$ absolute and $L_2$ relative errors remain consistent with other baselines. This indicates that a Transformer can maintain ICL capability even when trained with PINN-prior data. Also, this finding confirms that the model can effectively infer solutions from limited observed data $\tilde{D}$, even in the presence of inaccurate PINN-priors.

Table 3: Evaluation for convection, diffusion, and reaction equations measured at seen parameters, where the parameter values range from 1 to 20. Results are provided for each PINN-Prior Ratio, and for comparison, the results of baseline models are also included.

| Model | | Error Metric | convection $\beta \in [1, 20] \cap \mathbb{Z}$ | diffusion $\nu \in [1, 20] \cap \mathbb{Z}$ | reaction $\rho \in [1, 20] \cap \mathbb{Z}$ |
|---|---|---|---|---|---|
| Ours | 0% | Abs. err | 0.0764 | 0.0095 | 0.0160 |
| | | Rel. err | 0.0864 | 0.0134 | 0.0322 |
| | 20% | Abs. err | 0.1197 | 0.0088 | 0.0169 |
| | | Rel. err | 0.1276 | 0.0148 | 0.0390 |
| | 40% | Abs. err | 0.1543 | 0.0103 | 0.0286 |
| | | Rel. err | 0.1582 | 0.0149 | 0.0677 |
| | 60% | Abs. err | 0.1743 | 0.0172 | 0.0267 |
| | | Rel. err | 0.1746 | 0.0208 | 0.0679 |
| | 80% | Abs. err | 0.1677 | 0.0217 | 0.0327 |
| | | Rel. err | 0.1713 | 0.0300 | 0.0970 |
| | 100% | Abs. err | 0.1563 | 0.0200 | 0.0362 |
| | | Rel. err | 0.1654 | 0.0265 | 0.1136 |
| Prior Loss | | Abs. err | 0.0439 | 0.1262 | 0.0215 |
| | | Rel. err | 0.0441 | 0.1444 | 0.0845 |
| Hyper-LR-PINN | | Abs. err | 0.0340 | 0.1722 | 0.7111 |
| | | Rel. err | 0.0368 | 0.1991 | 0.7650 |
| $P^2$INN | | Abs. err | 0.2742 | 0.4553 | 0.1490 |
| | | Rel. err | 0.2743 | 0.5166 | 0.2915 |

**Remark 4.1.** *Notably, in diffusion case, the prior exhibits an error of $14\%$, yet our prediction error stands at just $2.6\%$. This discrepancy not only highlights the Transformer model's strong ICL capability but also demonstrates a form of superconvergence, where the model significantly outperforms expectations given the inaccurate prior. Such a result underscores the robustness and adaptability of our approach, reinforcing the idea that even with flawed prior information, the Transformer can extract meaningful insights and achieve high accuracy in predictions.*

### 4.2.3 TIME DOMAIN INTERPOLATION FOR UNSEEN PDE PARAMETERS

From this point, we train our model using only PINN-priors and further explore the base characteristics of SFMs'. In this section, we test our model with unseen parameters at convection, diffusion, and reaction systems. For each system, the model is trained with $[1, 20] \cap \mathbb{Z}$ range coefficients and tested with unseen coefficient $1.5, 2.5, \cdots, 19.5$ which is included in interval $[1, 20]$ and $20.5, 21.5, 22.5, \cdots, 30.5$ which is not in range of $[1, 20]$. The $L_2$ relative error measured for each coefficient value is plotted in Figure 3, along with the baselines Hyper-LR-PINN and the non-fine-tuned $P^2$INN.

Over the trained coefficient range, our model effectively interpolates the coefficients $\beta$, $\nu$, and $\rho$, achieving performance comparable to that seen with known coefficients. Moreover, the model demonstrates stable extrapolation in diffusion and reaction systems. Compared to the baselines, our model significantly outperforms it, particularly in diffusion and reaction systems. This result indicates that the Transformer can effectively learn the PPD of the prior space $\mathcal{D}$, even without observing the complete prior.

### 4.2.4 TIME DOMAIN EXTRAPOLATION FOR SEEN PDE PARAMETERS

One major limitation of the PINN is an extrapolation at the temporal domain that infer solutions at unknown points. Our model demonstrates extrapolation capability in the 1D convection equation, where the solution exhibits wave-like fluctuations in the inference region. In particular, the model trained with the PINN-prior $\mathcal{D}$ over the coefficient range $\beta \in [1, 20] \cap \mathbb{Z}$ can predict $\beta$ values in $1.5, 2.5, \cdots, 16.5$ for equations where the test points $\widetilde{T}$ fall within $t \in (0.6, 1.0]$, even though $\widetilde{D}$

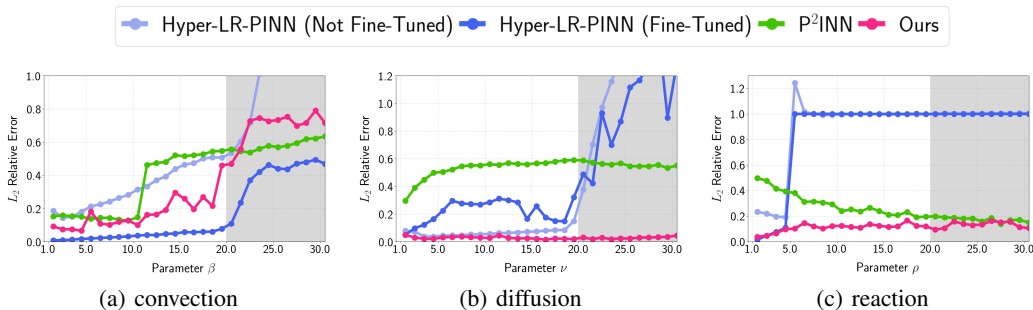

(a) convection     (b) diffusion     (c) reaction

Figure 3: The $L_2$ relative error measured at unseen parameters is presented for (a) convection, (b) diffusion, and (c) reaction, comparing our model with baseline methods. For Hyper-LR-PINN, both fine-tuned and non-fine-tuned results are plotted together. The grey area indicates the region where the model extrapolates the coefficient $\beta$, $\nu$, or $\rho$.

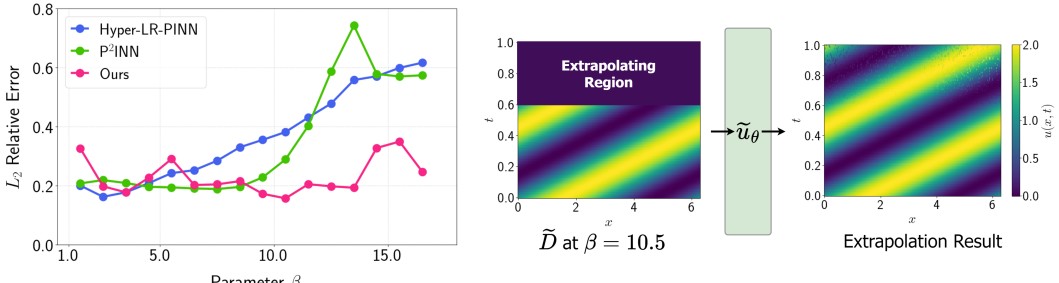

Figure 4: *(Left)* The $L_2$ relative error is evaluated for each convection coefficient $\beta = 1.5, 2.5, \cdots, 16.5$ as an extrapolation task. *(Right)* The graph illustrates the extrapolation of convection equation with $\beta = 10.5$ at $0.6 \le t \le 1.0$.

is only distributed within $t \in [0.0, 0.6]$. We then evaluate the $L_2$ relative error and plot for each coefficient $\beta$ with our baselines. Both baselines are not fine-tuned for each test $\beta$ to make a fair comparison with our zero-shot model.

As a result, our model demonstrates effective extrapolation capabilities in convection equation (Figure 4, *Left*). In addition, our model outperforms both Hyper-LR-PINN and P²INN across most values of $\beta$, while maintaining a stable $L_2$ relative error over a wider range. The diagram in Figure 4, *Right* presents the detailed performance at $\beta = 10.5$. This capability emphasizes our model's potential for advancing solutions to PDEs in unknown spatial regions and for enhancing time series predictions.

### 4.3 COMPREHENSIVE STUDY WITH VARIOUS REACTION TERMS

In this section, we expand the parameter space to following $\Omega$ using three different reaction terms: Fisher ($f_1$), Allen-Cahn ($f_2$), and Zeldovich ($f_3$),

$$u_t = \mathcal{N}(\cdot), \quad \mathcal{N}(t, x, u, \alpha) = -\beta u_x + \nu u_{xx} + \sum_{j=1}^{3} \rho_j f_j(u),$$

$$f_1 := u(1-u), \quad f_2 := u(1-u^2), \quad f_3 := u^2(1-u),$$
$$\Omega = \{\alpha := (\beta, \nu, \rho_1, \rho_2, \rho_3)\}. \tag{13}$$

To justify the expansion, we train the Transformer with $\beta = 0$, $\nu = 0$, and $\rho_j \in [1, 5] \cap \mathbb{Z}$ for $j = 1, 2, 3$ to evaluate its ICL capability in handling linear combinations of the reaction terms. The model is then tasked with inferring the solutions of the PDEs $u_t = \rho_1 f_1$, $u_t = \rho_2 f_2$, and $u_t = \rho_3 f_3$ to test whether it can generalize to CDR with unseen parameters and accurately distinguish between each component.

According to the result at Table 4, the $L_1$ absolute and $L_2$ relative errors are comparable to those obtained when trained with $\rho_1 \in [1, 5] \cap \mathbb{Z}$, suggesting the potential for expanding the parameter

Table 4: The Transformer is trained using a linear combination of Fisher, Allen-Cahn, and Zeldovich reaction terms with a given train parameter range. The $L_1$ absolute and $L_2$ relative errors for inferring each reaction term are then averaged over the given test parameter range. For comparison, the results for the Fisher reaction term tested in Section 4.2.1 are also included.

| Train Parameter Range | $\rho_1 \in [1,5] \cap \mathbb{Z}$ | $\rho_1, \rho_2, \rho_3 \in [1,5] \cap \mathbb{Z}$ | | |
|---|---|---|---|---|
| Test Parameter Range | $\rho_1 \in [1,5] \cap \mathbb{Z}$ | $\rho_1 \in [1,5] \cap \mathbb{Z}$ | $\rho_2 \in [1,5] \cap \mathbb{Z}$ | $\rho_3 \in [1,5] \cap \mathbb{Z}$ |
| $L_1$ Abs Err. | 0.0102 | 0.0755 | 0.0381 | 0.0611 |
| $L_2$ Rel Err. | 0.0154 | 0.1098 | 0.0734 | 0.0830 |

space. Specifically, the results demonstrate that our model can accurately distinguish between the three different reaction terms, even when trained with their linear combination. Although the model was trained using arbitrary linear combinations of terms commonly found in real-world applications, it is capable of effectively solving PDEs composed of meaningful combinations of these terms during testing. This demonstrates the model's ability to generalize beyond its training data and infer significant governing relationships from complex systems.

## 5 RELATED WORKS

**In-context learning**   Transformers have shown remarkable ICL abilities across various studies. They can generalize to unseen tasks by emulating Bayesian predictors (Panwar et al., 2024) and linear models (Zhang et al., 2024), while also efficiently performing Bayesian inference through Prior-Data Fitted Networks (PFNs) (Müller et al., 2021). Their robustness extends to learning different classes of functions, such as linear and sparse linear functions, decision trees, and two-layer neural networks even under distribution shifts (Garg et al., 2022). Furthermore, Transformers can adaptively select algorithms based on input sequences, achieving near-optimal performance on tasks like noisy linear models (Bai et al., 2023). They are also highly effective and fast for tabular data classification (Hollmann et al., 2022).

**Foundation model**   Recent studies have advanced in-context operator learning and PDE solving through Transformer-based models. Ye et al. (2024) introduces PDEformer, a versatile model for solving 1D PDEs with high accuracy and strong performance in inverse problems. In-context operator learning has also been extended to multi-modal frameworks, as seen in Yang et al. (2023c), where ICON-LM integrates natural language and equations to outperform traditional models. Additionally, Yang & Osher (2024) and Yang et al. (2023b) demonstrate the generalization capabilities of In-Context Operator Networks (ICON) in solving various PDE-related tasks, highlighting ICON's adaptability and potential for few-shot learning across different differential equation problems. Several other studies have addressed the problem of solving various PDEs using a single trained model (Hang et al., 2024; Herde et al., 2024) . However, many of these approaches rely on symbolic PDE information, true or near-true solutions and/or do not support zero-shot in-context learning, making their objectives different from ours.

## 6 CONCLUSION AND LIMITATIONS

In this work, we presented MaD-Scientist for scientific machine learning that integrates in-context learning and Bayesian inference for predicting PDE solutions. Our results demonstrate that Transformers, equipped with self-attention and cross-attention mechanisms, can effectively generalize from prior data, even in the presence of noise, and exhibit robust zero-shot learning capabilities. These findings suggest that foundation models in SciML have the potential to follow the development trajectory similar to that of natural language processing foundation models, offering new avenues for further exploration and advancement in the field.

The Transformer used in our study clearly demonstrates the ICL capability, when trained with PINN-based prior. However, it is limited to the CDR equations in our paper. We will consider other types of PDE and more diverse initial and boundary conditions in the future, enhancing its adaptability to real-world scenarios and its role as a foundation model.

ETHICS STATEMENT

This research adheres to the ethical standards required for scientific inquiry. We have considered the potential societal impacts of our work and have found no clear negative implications. All experiments were conducted in compliance with relevant laws and ethical guidelines, ensuring the integrity of our findings. We are committed to transparency and reproducibility in our research processes.

REPRODUCIBILITY

We are committed to ensuring the reproducibility of our research. All experimental procedures, data sources, and algorithms used in this study are clearly documented in the paper. The code and datasets will be provided as the supplementary material and be made publicly available upon publication, allowing others to validate our findings and build upon our work.

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

# A    THE PROOF OF THEOREM 2.1

*Proof.* For any $n, \epsilon$, we derive that

$$
\mathbb{E}_x \left[ H \left( p_{\hat{\theta}}(\cdot \mid x, D_n), \pi(\cdot \mid x) \right) \right] \leq \mathbb{E}_x \left[ H \left( p_{\hat{\theta}}(\cdot \mid x, D_n), \pi(\cdot \mid x, D_n) \right) \right] \tag{1}
$$
$$
+ \mathbb{E}_x \left[ H \left( \pi(\cdot \mid x, D_n), \pi(\cdot \mid x) \right) \right]
$$

$$
\leq \sqrt{ \frac{1}{2} \mathbb{E}_x \left[ KL \left( p_{\hat{\theta}}(\cdot \mid x, D_n), \pi(\cdot \mid x, D_n) \right) \right] } \tag{2}
$$
$$
+ \mathbb{E}_x \left[ H \left( \pi(\cdot \mid x, D_n), \pi(\cdot \mid x) \right) \right]
$$

$$
\leq \sqrt{\frac{\epsilon}{2}} + \mathbb{E}_x \left[ 1 - \int_{\mathcal{Y}} \sqrt{ \int q(y \mid x) \pi(y \mid x) \mathrm{d}\Pi^n(q) \mathrm{d}y } \right]^{1/2} \tag{3}
$$

$$
\leq \sqrt{\frac{\epsilon}{2}} + \left[ 1 - \int_{\mathcal{X}} \pi(x) \int_{\mathcal{Y}} \frac{1}{\pi(x)} \sqrt{ \int q(y, x) \pi(y, x) \mathrm{d}\Pi^n(q) \mathrm{d}y \mathrm{d}x } \right]^{1/2}
$$

$$
\leq \sqrt{\frac{\epsilon}{2}} + \left[ 1 - \int_{\mathcal{X}} \int_{\mathcal{Y}} \int \sqrt{ q(y, x) \pi(y, x) } \mathrm{d}\Pi^n(q) \mathrm{d}y \mathrm{d}x \right]^{1/2}
$$

$$
= \sqrt{\frac{\epsilon}{2}} + \left[ \int H \left( q, \pi \right)^2 \mathrm{d}\Pi^n(q) \right]^{1/2}
$$

$$
\leq \sqrt{\frac{\epsilon}{2}} + \left[ \int H \left( q, \pi \right) \mathrm{d}\Pi^n(q) \right]^{1/2}
$$

$$
= \sqrt{\frac{\epsilon}{2}} + \left[ \int_{\{q : H(\pi, q) > \epsilon\}} H \left( q, \pi \right) \mathrm{d}\Pi^n(q) \right]^{1/2}
$$

$$
+ \left[ \int_{\{q : H(\pi, q) \leq \epsilon\}} H \left( q, \pi \right) \mathrm{d}\Pi^n(q) \right]^{1/2} \tag{4}
$$

$$
= \sqrt{\frac{\epsilon}{2}} + \left( \Pi^n(\{q : H(\pi, q) > \epsilon\}) + \epsilon \right)^{1/2} \to \sqrt{\frac{\epsilon}{2}} + \sqrt{\epsilon} \quad \text{a.s.}
$$
$$
\tag{5}
$$

The first inequality (1) is derived from the triangle inequality for the Hellinger distance, which states that for any intermediate distribution $q(\cdot \mid x, D_n)$, we have

$$
H \left( p_{\hat{\theta}}(\cdot \mid x, D_n), \pi(\cdot \mid x) \right) \leq H \left( p_{\hat{\theta}}(\cdot \mid x, D_n), q(\cdot \mid x, D_n) \right) + H \left( q(\cdot \mid x, D_n), \pi(\cdot \mid x) \right).
$$

The second inequality (2) uses the fact that the Hellinger distance $H(p, q)$ is bounded above by the square root of the KL divergence $KL(p \parallel q)$, such that

$$
H(p, q)^2 \leq \frac{1}{2} KL(p \parallel q).
$$

Thus, we can bound the Hellinger distance by the KL divergence. In the third inequality (3), we make use of assumption

$$
\mathbb{E}_x \left[ KL \left( p_{\hat{\theta}}(\cdot \mid x, D_n), \pi(\cdot \mid x, D_n) \right) \right] < \epsilon,
$$

and utilize the definition of the Hellinger distance. In (4), we partition the domain into two regions– one where the Hellinger distance $H(\pi, q)$ exceeds $\epsilon$ and another where it is less than or equal to $\epsilon$–and use this partitioning to demonstrate the inequality.

Finally, in (5), by posterior consistency, the region where the Hellinger distance is greater than $\epsilon$ vanishes as $n \to \infty$ such that

$$
\Pi^n \left\{ q : H(\pi, q) > \epsilon \right\} \to 0 \quad \text{almost surely.}
$$

Since $\epsilon$ is arbitrary, we can conclude that

$$
\mathbb{E}_x \left[ H \left( p_{\hat{\theta}}(\cdot \mid x, D_n), \pi(\cdot \mid x) \right) \right] \xrightarrow{n \to \infty} 0 \quad \text{almost surely.}
$$

$\square$

## B    ENVIRONMENTS

We conducted the experiments using Python 3.8.19, PyTorch 2.4.0+cu121, scikit-learn 1.3.2, NumPy 1.24.4, and pandas 2.0.3, with CUDA 12.1, NVIDIA Driver 535.183.01, and an NVIDIA RTX A6000. Additionally, we used CUDA 12.4, NVIDIA Driver 550.67, and either an NVIDIA GeForce RTX 3090 or an NVIDIA TITAN RTX.

## C    COMPARISON BETWEEN BASELINES

In addition to the comparison between baselines in 1, the additional comparisons between baselines is shown below in Table 5. For Hyper-LR-PINN, the results are estimated solely for Phase 1 and the number of model parameters estimated refers specifically to the model used in sections 4.2.

Table 5: Additional comparisons between baselines and our model in 1D CDR experiments.

| Properties | Hyper-LR-PINN | $P^2$INN | Ours |
|---|---|---|---|
| Number of model parameters | 28,151 | 76,851 | 21,697 |
| Training time per epoch(s) | 1.8613 | 0.9640 | 0.1187 |
| GPU memory usage(MB) | 1,175 | 1,108 | 998 |

For the 2D CDR experiments, Table 6 shows the training time per epoch, number of parameters and GPU memory usage for our model and the baselines in their best settings.

Table 6: Additional comparisons between baselines and our model in 2D CDR experiments.

| Task | Model | Number of model parameters | Training time per epoch(s) | GPU memory usage(MB) |
|---|---|---|---|---|
| Interpolation with numerical prior | Ours | 30,273 | 1.6869 | 12,435.89 |
| | DeepONet | 650,113 | 0.5742 | 25.70 |
| Extrapolation with numerical prior | Ours | 60,065 | 1.5077 | 7,536.04 |
| | A-FNO | 385,728 | 1.6085 | 21.29 |
| | F-FNO | 173,063,937 | 0.8825 | 2,080.55 |
| | FNO | 8,980,929 | 1.5927 | 196.18 |
| | DeepONet | 21,377 | 0.7818 | 28.76 |
| Interpolation with PINN prior | Ours | 30,273 | 1.6526 | 12,435.91 |
| | DeepONet | 1,357,697 | 0.5941 | 43.04 |
| Extrapolation with PINN prior | Ours | 47,361 | 1.5072 | 9,309.12 |
| | A-FNO | 687,296 | 1.6755 | 25.46 |
| | F-FNO | 173,063,937 | 0.8847 | 2,080.55 |
| | FNO | 8,980,929 | 1.6135 | 196.18 |
| | DeepONet | 25,537 | 0.7854 | 31.53 |

## D    DATASETS

The specific information about PDE types used in the study is following. $n$ represent the maximum values for each parameter $\beta, \nu, \rho_1, \rho_2$ and $\rho_3$'s range, which are set to 5, 10, and 20 in our study.

For each specific PDE, we collect 256 initial points, 1,000 collocation points, 100 boundary points, and 1,000 test points. For each 1,000 collocation points, we sample 800 points for $D \cup T$ and 200 points for $\tilde{D}$ which are not overlapped. During the training phase, 30% of the data points are designated as $D$, while the remaining 70% are allocated to $T$.

## E    PINN USED IN PRIOR GENERATION

In this study, we utilize the PINN introduced by Raissi et al. (2019) to generate PINN-priors. The loss function employed during the training of the PINN is as follows:

$$\mathcal{L} = \mathcal{L}_u + \mathcal{L}_f + \mathcal{L}_b, \tag{14}$$

Table 7: PDEs used in our study and corresponding dataset information for each section. "CDR" means "convection-diffusion-reaction."

| Section | Equation Type | Equation | Parameter Range | Number of Datasets |
|---|---|---|---|---|
| 4.2 | convection | $u_t = -\beta u_x$ | $\beta \in [1, n] \cap \mathbb{Z}$ | $n$ |
| | diffusion | $u_t = \nu u_{xx}$ | $\nu \in [1, n] \cap \mathbb{Z}$ | $n$ |
| | reaction | $u_t = \rho_1 u(1 - u)$ | $\rho_1 \in [1, n] \cap \mathbb{Z}$ | $n$ |
| | convection-diffusion | $u_t = -\beta u_x + \nu u_{xx}$ | $\beta, \nu \in [1, n] \cap \mathbb{Z}$ | $n^2$ |
| | reaction-diffusion | $u_t = \nu u_{xx} + \rho_1 u(1 - u)$ | $\nu, \rho_1 \in [1, n] \cap \mathbb{Z}$ | $n^2$ |
| | CDR | $u_t = -\beta u_x + \nu u_{xx} + \rho_1 u(1 - u)$ | $\beta, \nu, \rho_1 \in [1, n] \cap \mathbb{Z}$ | $n^3$ |
| 4.3 | Table 4 | $u_t = \rho_1 u(1-u) + \rho_2 u(1-u^2) + \rho_3 u^2(1 - u)$ | $\rho_1, \rho_2, \rho_3 \in [1, n] \cap \mathbb{Z}$ | $n^3$ |

where $\mathcal{L}_u, \mathcal{L}_f$ and $\mathcal{L}_b$ is defined as

$$\mathcal{L}_u = \frac{1}{N_u} \sum (\tilde{u}(x, 0) - u(x, 0))^2, \quad \mathcal{L}_f = \frac{1}{N_f} \sum (\mathcal{N}(t, x, u, \alpha))^2, \quad \mathcal{L}_b = \frac{1}{N_b} \sum (\tilde{u}(0, t) - \tilde{u}(2\pi, t))^2,$$

$$(15)$$

for $N_u$ points at initial condition, $N_f$ collocation points and $N_b$ boundary points.

# F TRAINING ALGORITHM

We train the Transformer like following.

---
**Algorithm 1** Training a Transformer
---
1: **Input:** A prior dataset $D \cup T$ drawn from prior $p(\mathcal{D})$
2: **Output:** A Transformer $\tilde{u}_\theta$ which can approximate the PPD
3: Initialize the Transformer $\tilde{u}_\theta$
4: **for** $i = 1$ to $n$ **do**
5:      Sample $\alpha \in \Omega$ and $D \cup T \subseteq \tilde{u}(\alpha) \sim p(\mathcal{D})$
6:      $(D := \{(x_D^{(i)}, t_D^{(i)})\}_{i=1}^{N_D}, T := \{(x_T^{(j)}, t_T^{(j)})\}_{j=1}^{N_T})$
7:      Compute loss $L_\alpha = \frac{1}{N_T} \sum_{j=1}^{N_T} \left\{ \tilde{u}_\theta(x_T^{(j)}, t_T^{(j)} | D_n) - \tilde{u}(x_T^{(j)}, t_T^{(j)}) \right\}^2$.
8:      Update parameters $\theta$ with an Adam optimizer
9: **end for**
---

# G ICL OF TRANSFORMERS WITH NOISY PRIOR

In order to study the ICL capability of Transformers with noisy prior, we introduce four kinds of prior $\mathcal{D}$ like

**P1** (noiseless) : $p(\mathcal{D}) = p(\mathcal{U})$,          **P2** (Gaussian noise) : $p(\mathcal{D}) \sim \mathcal{N}(\mathcal{U}, \sigma^2 \mathbf{I})$,

**P3** (salt-and-pepper noise) : $p(\mathcal{D}) \sim p(s \cdot \mathcal{U})$ where $s = \begin{cases} \min(\mathcal{U}) & \text{with probability } \frac{\gamma}{2}, \\ \max(\mathcal{U}) & \text{with probability } \frac{\gamma}{2}, \\ 1 & \text{with probability } 1 - \gamma, \end{cases}$

**P4** (uniform noise) : $p(\mathcal{D}) \sim p(\mathcal{U} + U(-\epsilon, \epsilon))$ ($U$: uniform distribution).
We sample $D \cup T \sim p(\mathcal{D})$, where $\mathcal{D}$ is a noisy prior, and train the Transformer $\tilde{u}_\theta$. We then test $\tilde{u}_\theta$ with $\tilde{D} \cup \tilde{T} \sim p(\mathcal{U})$, demonstrating that the model can predict the true solution even when trained on noisy prior data. The experiment is conducted on reaction and convection-diffusion-reaction equations, which outperform other baselines, under three different noises: the Gaussian noise (**P2**), the salt-and-pepper noise (**P3**), and the uniform noise (**P4**). The standard deviation $\sigma$ of Gaussian noise is set to 1%, 5%, and 10% of the mean value of the ground truth solution. Additionally, for

Table 8: The $L_1$ absolute and $L_2$ relative errors for the reaction and convection-diffusion-reaction systems using the **P2** prior with varying levels of Gaussian noise $\sigma$, **P3** prior with varying levels of noise probe $\gamma$, and **P4** prior with varying levels of noise $\epsilon$ (1%, 5%, and 10%). For a comparison, the result of using **P1** prior is notated.

| System | Prior Type | Noisy Prior with a Noise Level | | | | | | P1 Prior | |
|---|---|---|---|---|---|---|---|---|---|
| | | 1% Noise | | 5% Noise | | 10% Noise | | | |
| | | Abs. err | Rel. err | Abs. err | Rel. err | Abs. err | Rel. err | Abs. err | Rel. err |
| Reaction | P2 | 0.0210 | 0.0392 | 0.0213 | 0.0399 | 0.0210 | 0.0392 | 0.0160 | 0.0322 |
| | P3 | 0.0309 | 0.0598 | 0.0286 | 0.0517 | 0.0354 | 0.0619 | | |
| | P4 | 0.0285 | 0.0568 | 0.0293 | 0.0583 | 0.0306 | 0.0607 | | |
| Convection -Diffusion -Reaction | P2 | 0.0175 | 0.0296 | 0.0220 | 0.0431 | 0.0235 | 0.0431 | 0.0159 | 0.0310 |
| | P3 | 0.0246 | 0.0459 | 0.0263 | 0.0453 | 0.0267 | 0.0496 | | |
| | P4 | 0.0210 | 0.0420 | 0.0215 | 0.0422 | 0.0230 | 0.0426 | | |

the experiment, the probe $\gamma$ for salt-and-pepper noise and the range $\epsilon$ for uniform noise are also set to 1%, 5%, and 10%.

Our model demonstrates robust performance across different types of noise injection as shown in Table 8. It shows our Transformer can perform ICL with zero-shot learning even if it is trained with inaccurate or noisy prior $D \cup T \sim p(\mathcal{D})$.

## H  EXPERIMENTS AT PINN FAILURE MODES

Referring to Cho et al. (2024) and Krishnapriyan et al. (2021), we test our method on PINN's major failure modes: $\beta \in [30, 40]$ with an initial condition $1 + \sin(x)$ and $\rho \in [1, 10]$ with an initial condition $\mathcal{N}\left(\pi, \left(\frac{\pi}{2}\right)^2\right)$. We have trained our model with this range with **P1** prior and evaluate $L_1$ absolute and $L_2$ relative errors. The following are major results and solution profiles at failure modes.

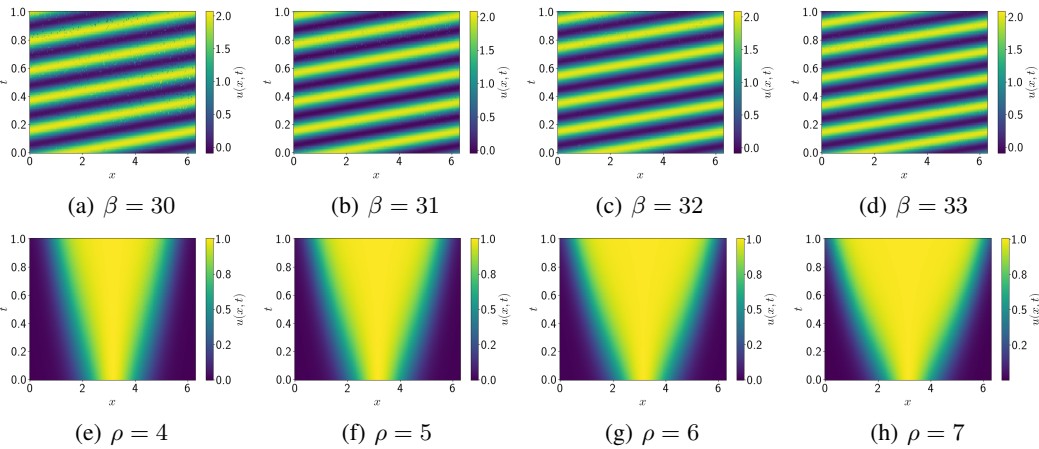

(a) $\beta = 30$      (b) $\beta = 31$      (c) $\beta = 32$      (d) $\beta = 33$

(e) $\rho = 4$      (f) $\rho = 5$      (g) $\rho = 6$      (h) $\rho = 7$

Figure 5: The solution profiles at PINN failure modes: (a), (b), (c) and (d) for $\beta \in [30, 40]$ with initial condition $1 + \sin(x)$ and (e), (f), (g) and (h) for $\rho \in [1, 10]$ with initial condition $\mathcal{N}\left(\pi, \left(\frac{\pi}{2}\right)^2\right)$. The solution profile is constructed using the union of 1,000 test prediction points and the remaining ground truth points.

Table 9: The $L_1$ absolute and $L_2$ relative error at PINN failure modes.

| Trained Coefficient Range | Test Coefficient Value | $L_2$ Error Type | | Average Error | |
|---|---|---|---|---|---|
| | | Abs.err | Rel.err | Abs.err | Rel.err |
| $\beta \in [30, 40]$ | $\beta = 30$ | 0.2483 | 0.2516 | 0.1280 | 0.1328 |
| | $\beta = 31$ | 0.1029 | 0.1111 | | |
| | $\beta = 32$ | 0.0803 | 0.0882 | | |
| | $\beta = 33$ | 0.0806 | 0.0801 | | |
| $\rho \in [1, 10]$ | $\rho = 4$ | 0.0071 | 0.0160 | 0.0048 | 0.0097 |
| | $\rho = 5$ | 0.0029 | 0.0054 | | |
| | $\rho = 6$ | 0.0033 | 0.0063 | | |
| | $\rho = 7$ | 0.0058 | 0.0112 | | |

# I  ADDITIONAL EXPERIMENTS ON 2D CDR

In this section, we extend our evaluation to the two-dimensional convection-diffusion-reaction (2D CDR) equation, which represents a more complex and higher-dimensional setting compared to the 1D CDR experiments discussed earlier. The goal of this experiment is to evaluate how well our model handles higher-dimensional PDEs and compare its performance with widely used neural operator methods in both interpolation and extrapolation tasks. The following two-dimensional convection-diffusion-reaction (CDR) equation is used for the benchmark PDE,

$$\text{2D CDR: } u_t + \beta_x u_x + \beta_y u_y - \nu u_{xx} - \nu_y u_{yy} - \rho u(1 - u) = 0,$$
$$, x \in [0, \pi], \quad y \in [0, \pi], \quad t \in [0, 0.7], \quad \beta_x, \beta_y, \nu_x, \nu_y, \rho \in [1, 3] \cap \mathbb{Z}. \tag{16}$$

We only consider the equations with all non-zero coefficients in 2D CDR experiments.

## I.1  EXPERIMENTAL SETUP AND BASELINES

For the 2D CDR experiments, we compare our model with neural operator-based baselines: Deep-ONet, FNO, F-FNO, and A-FNO. These baselines are selected for their scalability and ability to handle high-dimensional PDEs effectively. Due to memory constraints, baselines like Hyper-LR-PINN and P$^2$INN from the 1D CDR experiments are excluded. The detailed hyperparameters, training and testing settings, and computational costs used in the 2D CDR experiments are described thoroughly in Appendix N.

**Baseline methods**  We compare our model with 4 baselines: deep operator network (Deep-ONet) (Lu et al., 2021), Fourier Neural Operator(FNO) (Li et al., 2020), Factorized Fourier Neural Operator(F-FNO) (Tran et al., 2021), and Adaptive Fourier Neural Operator(A-FNO) (Guibas et al., 2021). These models are based on neural operator.

- DeepONet is a neural operator architecture designed to learn operators mapping input functions to output functions. It combines branch and trunk networks to predict values in a function space.
- FNO learns solution operators for partial differential equations using the Fourier transform. By mapping inputs to a frequency domain, FNO captures complex patterns and long-range dependencies and models complex systems.
- F-FNO extends the Fourier Neural Operator by factorizing its layers to reduce computational costs. This factorization enables efficient learning of solution operators for complex systems.
- A-FNO is a variant of the Fourier Neural Operator that dynamically adjusts the resolution of the frequency domain during training. This adaptation aims to capture relevant features across scales, enabling more flexible modeling of complex systems.

**Interpolation Task**  DeepONet is the sole baseline used for the interpolation task. FNO, F-FNO, and A-FNO were specifically designed to take grid input at a specific time point and predict grid output subsequent time steps. As these models were originally developed for time trajectory prediction

tasks, they are not suitable for interpolation. Consequently, DeepONet serves as the sole baseline for comparison in this task.

**Extrapolation Task**   All baselines, including DeepONet, FNO, F-FNO, and A-FNO, are evaluated to compare their rollout performance in predicting future time steps.

## I.2   TIME DOMAIN INTERPOLATION FOR SEEN PDE PARAMETERS WITH A NUMERICAL PRIOR

We first conduct an experiment on time domain interpolation for seen PDE Parameters with a numerical prior. Our model follows the same experimental setup as in Section 4.2.1, while using the increased number of data points. For DeepONet, we implement a structure where it takes initial and boundary points as inputs and predicts data points. Table 10 shows the evaluation results measured by $L_1$ absolute error, $L_2$ relative error, and $L_\infty$ relative error. Additionally, we evaluate on another test dataset in grid format for the $H_1$ norm metric, and the corresponding results have been included as well.

Table 10: Evaluation for interpolation task in 2D CDR equation with Fisher reaction term using numerical prior. They are measured at seen parameters, where the parameter values are $\beta_x, \beta_y, \nu_x, \nu_y, \rho \in [1,3] \cap \mathbb{Z}$. Best performance is marked in **bold**.

| Metric | Ours | DeepOnet |
|---|---|---|
| $L_1$ Abs.error | **0.00584** | 0.00763 |
| $L_2$ Rel.error | **0.01185** | 0.01368 |
| $L_\infty$ Rel.error | 0.05862 | **0.04533** |
| $H_1$ Abs.error | **0.04885** | 0.11168 |

As a result, our model outperforms DeepONet on $L_1$ absolute error, and $L_2$ relative error, and $H_1$ absolute error. We can confirm that our model maintains its performance in the 2D CDR interpolation task based on the numerical prior when extended from the 1D CDR to higher dimensions.

## I.3   TIME DOMAIN EXTRAPOLATION FOR SEEN PDE PARAMETERS WITH A NUMERICAL PRIOR

In this section, we conduct an experiment on time domain extrapolation for seen PDE Parameters with a numerical prior. The training data $D$ and $T$ are randomly distributed over $t \in [0.0, 0.7]$. During the test phase, grid data $\widetilde{D}$ at $t = 0.7$ is provided as input to predict the values of grid points $\widetilde{T}$ at $t = 0.8, 0.9, 1.0$. For the baselines, the models are trained to take grid data $D$ at a specific time point as input and predict the grid point values $T$ at the next time step. During testing, the grid data $\widetilde{D}$ at $t = 0.7$ is provided as input, and the models conduct rollout to predict the grid point values $\widetilde{T}$ at $t = 0.8, 0.9, 1.0$. Table 11 shows the evaluation results measured by $L_1$ absolute error, $L_2$ relative error, and $L_\infty$ relative error.

Table 11: Evaluation for extrapolation task in 2D CDR equation with Fisher reaction term using numerical prior. They are measured at seen parameters, where the parameter values are $\beta_x, \beta_y, \nu_x, \nu_y, \rho \in [1,3] \cap \mathbb{Z}$. Best performance is marked in **bold**.

| Metric | Ours | A-FNO | F-FNO | FNO | DeepOnet |
|---|---|---|---|---|---|
| $L_1$ Abs.error | **0.00115** | 0.00283 | 0.00284 | 0.00291 | 0.01530 |
| $L_2$ Rel.error | **0.00127** | 0.00353 | 0.00355 | 0.00361 | 0.02308 |
| $L_\infty$ Rel.error | **0.00261** | 0.00976 | 0.01230 | 0.01182 | 0.13671 |

As a result, our model surpasses all baseline models across the three evaluation metrics. These results indicate that our model successfully generalizes from the 1D CDR to the 2D CDR extrapolation task with the numerical prior.

### I.4 TIME DOMAIN INTERPOLATION FOR SEEN PDE PARAMETERS WITH A PINN-PRIOR

In this section, we explore time-domain interpolation for previously seen PDE parameters using a 100% PINN prior. Table 12 presents the quality of the PINN prior and the evaluation results, measured by $L_1$ absolute error, $L_2$ relative error, and $L_\infty$ relative error. The PINN loss indicates the degree of error between the PINN-prior data used for model training and the corresponding numerical values, providing a quantitative measure of the PINN-prior quality. Furthermore, we additionally evaluate on another test dataset in grid format for the $H_1$ norm metric, and the corresponding results have been included as well.

Table 12: PINN-prior quality and evaluation for interpolation task in 2D CDR equation with Fisher reaction term using 100% PINN-prior. They are measured at seen parameters, where the parameter values are $\beta_x, \beta_y, \nu_x, \nu_y, \rho \in [1, 3] \cap \mathbb{Z}$. Best performance is marked in **bold**.

| Metric | PINN loss | Ours | DeepOnet |
|---|---|---|---|
| $L_1$ Abs.error | 0.01225 | **0.00730** | 0.00818 |
| $L_2$ Rel.error | 0.01740 | **0.01319** | 0.01480 |
| $L_\infty$ Rel.error | 0.04980 | 0.05769 | **0.05171** |
| $H_1$ Abs.error | 0.11530 | **0.06488** | 0.09756 |

As a result, our model achieves superior performance compared to DeepONet in $L_1$ absolute error, and $L_2$ relative error, and $H_1$ absolute error. These results emphasize the capability of our model to excel in interpolation tasks, not only with the numerical prior (Table 10) but also with the PINN prior (Table 12), as it adapts and generalizes effectively while transitioning from the 1D CDR to higher-dimensional scenarios.

### I.5 TIME DOMAIN EXTRAPOLATION FOR SEEN PDE PARAMETERS WITH A PINN-PRIOR

In this section, we conduct an experiment on time domain extrapolation for seen PDE parameters with a 100% PINN-prior. Table 13 shows the PINN-prior quality and the evaluation results measured by $L_1$ absolute error, $L_2$ relative error, and $L_\infty$ relative error.

Table 13: PINN-prior quality and evaluation for extrapolation task in 2D CDR equation with Fisher reaction term using 100% PINN-prior. They are measured at seen parameters, where the parameter values are $\beta_x, \beta_y, \nu_x, \nu_y, \rho \in [1, 3] \cap \mathbb{Z}$. Best performance is marked in **bold**.

| Metric | PINN loss | Ours | A-FNO | F-FNO | FNO | DeepOnet |
|---|---|---|---|---|---|---|
| $L_1$ Abs.error | 0.00619 | **0.00296** | 0.00619 | 0.00978 | 0.00431 | 0.01150 |
| $L_2$ Rel.error | 0.00763 | **0.00328** | 0.00763 | 0.01217 | 0.00568 | 0.01495 |
| $L_\infty$ Rel.error | 0.00763 | **0.00702** | 0.01869 | 0.04701 | 0.2175 | 0.08218 |

As a result, our model outperforms other baselines across all three metrics. This result confirms that when extending from the 1D CDR to higher dimensions, our model maintains its performance in the extrapolation task not only with the numerical prior but also with the PINN prior.

## J ADDITIONAL EXPERIMENTS ON BIHARMONIC EQUATION

To demonstrate the performance of our model on PDEs involving higher-order derivatives, we conducted an additional experiment. Specifically, we evaluated the model's performance on an interpolation task using the Biharmonic equation, a commonly studied PDE that involves higher-order derivatives.

### J.1 EXPERIMENTAL SETUP AND BASELINES

The following two-dimensional biharmonic equation is used:

$$\text{2D biharmonic: } u_t + u_{xxxx} + 2u_{xxyy} + u_{yyyy} = 0, x \in [0, 2\pi], y \in [0, 2\pi], t \in [0, 0.5],$$
$$\text{Initial condition: } u(0, x, y) = \sin(x)\sin(y). \tag{17}$$

The two-dimensional biharmonic equation has an exact solution under certain initial conditions. Therefore, we adopt initial conditions with known exact solutions and use them as the analytical prior for both training and testing.

$$\text{Exact solution: } u(t, x, y) = e^{-t} \sin(x) \sin(y). \tag{18}$$

**Baseline methods**  We compare our model with deep operator network (DeepONet). DeepONet uses the same model architecture as in the 2D CDR interpolation task. Additionally, since this experiment focuses on the interpolation task, FNO, F-FNO, and A-FNO are excluded from the baselines. Hyper-LR-PINN and P$^2$INN, which are Parameterized PINNs designed for learning multiple PDEs, are also excluded as they are not suited for this specific experiment.

### J.2    Time Domain Interpolation with an Analytical Prior

We conduct an experiment on time domain interpolation for the biharmonic equation with an analytical prior. Our model and DeepONet follow the same experimental setup as in Appendix K.2. Table 14 shows the evaluation results measured by $L_1$ absolute error, $L_2$ relative error, and $L_\infty$ relative error.

Table 14: Evaluation for interpolation task in biharmonic equation using analytical prior. Best performance is marked in **bold**.

| Metric | Ours | DeepOnet |
|---|---|---|
| $L_1$ Abs.error | **0.01525** | 0.12730 |
| $L_2$ Rel.error | **0.05307** | 0.43996 |
| $L_\infty$ Rel.error | **0.08986** | 0.58891 |

As a result, our model outperforms DeepONet across all three metrics. We can confirm that our model maintains its performance in the biharmonic equation interpolation task based on the analytical prior, which involves higher-order derivatives.

## K    Additional Experiments on Burgers' Equation

To evaluate the performance of our model on PDEs involving stiffness, we conduct an additional experiment using the Burgers' equation. This equation is a well-known PDE that serves as a simplified model in various scientific and engineering fields, such as fluid dynamics and wave propagation. It captures key nonlinear phenomena, including advection and diffusion, making it a valuable benchmark for testing computational methods. When the viscosity parameter is small, the Burgers' equation can develop shock formations, which pose significant challenges for classical numerical methods. This characteristic makes it an ideal test case for assessing whether models can effectively handle such complexities.

### K.1    Experimental Setup and Baselines

In this experiment, we focus on time-direction extrapolation. We train the models on data in the domain prior to the appearance of the shock and then test on data in the domain after the shock occurs. Specifically, the model are trained on data within the time range $t \in [0.0, 0.25]$, and then tested on its ability to predict the solution at later time points, $t \in \{0.4, 0.5, 0.6\}$, given data at $t = 0.3$. The data used for training and testing are generated by a physics-informed neural network (PINN) (Raissi et al., 2019).

In one spatial dimension, the Burgers' equation with Dirichlet boundary conditions is expressed as:

$$
\begin{aligned}
&u_t + u u_x - \frac{0.01}{\pi} u_{xx} = 0, \quad x \in [-1, 1],\ t \in [0.0, 0.99], \\
&u(0, x) = -\sin(\pi x), \\
&u(t, -1) = u(t, 1).
\end{aligned}
\tag{19}
$$

**Baseline methods**  We compare our model with deep operator network (DeepONet) and Fourier Neural Operator (FNO). Since we conduct the experiment on 1D PDE, our model uses the same structure as in the 1D CDR extrapolation task. Also, we adapt the architectures of DeepONet and FNO, which were used in the 2D CDR extrapolation experiment, to suit the 1D problem. Factorized Fourier Neural Operator (F-FNO) and Adaptive Fourier Neural Operator (A-FNO), Hyper-LR-PINN, and $P^2$INN are also excluded as they are not suited for this specific experiment.

### K.2  TIME DOMAIN EXTRAPOLATION WITH AN ANALYTICAL PRIOR

We conduct an experiment on time domain interpolation for Burgers' equation with a numerical prior. Table 15 shows the evaluation results measured by $L_1$ absolute error, $L_2$ relative error, and $L_\infty$ relative error.

Table 15: Evaluation for extrapolation task in Burgers' equation using numerical prior. Best performance is marked in **bold**.

| Metric | Ours with PINN loss | Ours | FNO | DeepOnet |
|---|---|---|---|---|
| $L_1$ Abs.error | **0.03078** | 0.06367 | 0.06153 | 0.13612 |
| $L_2$ Rel.error | **0.15044** | 0.21786 | 0.15619 | 0.26794 |
| $L_\infty$ Rel.error | 0.73430 | 0.80187 | **0.62267** | 0.62590 |

The experimental results show that our model outperforms DeepONet in two metrics but performs worse than FNO across all three metrics. This indicates that our current model structure struggles to infer values for unexpected patterns, such as shocks, when trained on data from before the shock occurs.

To address this issue, we experiment with incorporating a PINN loss into our model during training. As a result, the model achieves better performance than the baselines on two metrics. This demonstrates the potential of leveraging PINN loss as a way to improve our model in future developments.

## L  EXTENSION OF EXPERIMENTS IN SECTION 4.3

In this section, we conducted additional experiments to further enhance the analysis of reaction equations with various reaction terms presented in Section 4.3. The experiments presented here incorporate fine-tuning, which was not used in our model that relied solely on pre-training through in-context learning (ICL). Through these experiments, we aim to demonstrate that while our model performs well with pre-training alone, fine-tuning can further improve its performance when higher accuracy is desired. Additionally, All experiments were conducted entirely based on PINN prior.

### L.1  FINE-TUNING FOR ADAPTING TO EACH REACTION SYSTEM

To enhance the performance of our model discussed in Section 4.3 by adapting to each reaction system, we perform fine-tuning on each target reaction equation. First, we performed pre-training on reaction equation having all reaction terms with nonzero coefficients $\rho_1, \rho_2, \rho_3 \in [1,5] \cap \mathbb{Z}$. Subsequently, we fine-tuned the model on target reaction equation containing only a single reaction term, $\rho_j \in [1,5] \cap \mathbb{Z}$ with $j = 1, 2, 3$, respectively. Afterward, we evaluated the model.

Table 16: Evaluation after fine-tuning the model with target test parameter range.

| Train Parameter Range | $\rho_1, \rho_2, \rho_3 \in [1,5] \cap \mathbb{Z}$ | | |
|---|---|---|---|
| **Test Parameter Range** | $\rho_1 \in [1,5] \cap \mathbb{Z}$ | $\rho_2 \in [1,5] \cap \mathbb{Z}$ | $\rho_3 \in [1,5] \cap \mathbb{Z}$ |
| $L_1$ **Abs Err.** | 0.0159 | 0.0142 | 0.0112 |
| $L_2$ **Rel Err.** | 0.0279 | 0.0333 | 0.0184 |
| $L_\infty$ **Rel Err.** | 0.0937 | 0.1407 | 0.0442 |

Table 17 shows the evaluation results measured by $L_1$ absolute error, $L_2$ relative error, and $L_\infty$ relative error. Experimental results demonstrate that the fine-tuning can improve our model's performance compared to the performance in Section 4.3.

## L.2 Expanding Parameter Space

In this section, we expand the parameter space of the pre-training dataset, originally defined in Section 4.3 as $\{(\beta, \nu, \rho_1, \rho_2, \rho_3) \mid \beta = \nu = 0, \rho_1, \rho_2, \rho_3 \in [1,5] \cap \mathbb{Z}\}$. Here, we generalize it to $\{(\beta, \nu, \rho_1, \rho_2, \rho_3) \mid \beta, \nu, \rho_1, \rho_2, \rho_3 \in [1,5] \cap \mathbb{Z}\}$, thereby including nonzero coefficients for $\beta$ and $\nu$. In other words, we conducted experiments on the 1D CDR equation with three reaction terms. Following the same procedure as outlined in Appendix L.1, pre-training was performed on 1D CDR equations with nonzero coefficients $\beta, \nu, \rho_1, \rho_2, \rho_3 \in [1,5] \cap \mathbb{Z}$. Subsequently, fine-tuning was carried out on target convection, diffusion, and three distinct reaction equations, followed by evaluation.

Table 17: Evaluation of training over the expanded parameter space followed by fine-tuning.

| Train Parameter Range | $\beta, \nu, \rho_1, \rho_2, \rho_3 \in [1,5] \cap \mathbb{Z}$ | | | | |
|---|---|---|---|---|---|
| **Test Parameter Range** | $\beta \in [1,5] \cap \mathbb{Z}$ | $\nu \in [1,5] \cap \mathbb{Z}$ | $\rho_1 \in [1,5] \cap \mathbb{Z}$ | $\rho_2 \in [1,5] \cap \mathbb{Z}$ | $\rho_3 \in [1,5] \cap \mathbb{Z}$ |
| $L_1$ **Abs Err.** | 0.0817 | 0.0418 | 0.0466 | 0.0334 | 0.0257 |
| $L_2$ **Rel Err.** | 0.0842 | 0.0477 | 0.0850 | 0.0883 | 0.0365 |
| $L_\infty$ **Rel Err.** | 0.1459 | 0.0731 | 0.2487 | 0.3294 | 0.0845 |

Table 17 presents the evaluation results, measured by $L_1$ absolute error, $L_2$ relative error, and $L_\infty$ relative error. The experimental results demonstrate that the model successfully distinguishes between five distinct equations, even when trained on an expanded parameter space. Additionally, the results highlight the model's robustness and adaptability to the expanded parameter space.

## M Sensitivity to The Number of Test Given Data Points

In this section, we provide empirical validation of Theorem 2.1, which establishes the theoretical consistency of the neural network's posterior predictive distribution (PPD) as the size of the data $D_n$ increases. Specifically, we evaluate the sensitivity of the neural network's PPD to the number of the given data $\widetilde{D}$ provided during the test process.

To this end, we conduct experiments by varying the number of the given data $\widetilde{D}$ while keeping other factors unchanged, including the best hyperparameter settings for the numerical prior-based interpolation task in 2D CDR. The results, depicted in Figure 6, clearly demonstrate that as the size of $\widetilde{D}$ increases, the error consistently decreases across all three evaluation metrics. This behavior aligns perfectly with the theoretical prediction in Theorem 2.1, where the posterior approximation is shown to converge toward the true distribution as the data size grows.

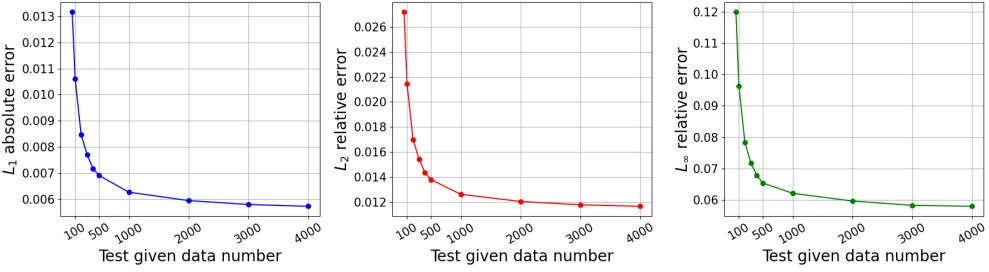

Figure 6: Sensitivity to the number of the given data $\widetilde{D}$ during test.

These experimental findings not only validate the theoretical insights of Theorem 2.1 but also emphasize the robustness and accuracy of the neural network's PPD approximation under the given prior. The decreasing error trend highlights how the model effectively integrates increasing amounts of data to produce predictions that are more consistent with the true underlying posterior distribution. This synergy between theory and empirical observation strongly supports the reliability and effectiveness of the proposed approach.

# N  EXPERIMENTAL DETAILS

In this section, we describe the hyperparameters and the number of training and testing data used in the experiments for each model. For the 1D CDR experiments in Section 4, we used the same number of data across all tasks when training or testing each equation. For the 2D CDR experiments in Appendix I, different numbers of data are used for interpolation and extrapolation task. Additionally, experiments using different priors within the same task are conducted under identical settings. For the 2D biharmonic experiment in Appendix J, we use fewer training data points compared to the 2D CDR interpolation task, as the experiment is conducted over a shorter time range. The number of data used for training and testing each task can be found in Table 18.

Table 18: Data number used in training and test of all experiments.

| Task | Model | Train data $D \cup T$ **number** | Test given data $\tilde{D}$ **number** | Test target points $\tilde{T}$ **number** |
|---|---|---|---|---|
| 1D CDR | Ours | 1,156 | 200 | 1,000 |
| | Hyper-LR-PINN | 1,156 | 200 | 1,000 |
| | P$^2$INN | 1,156 | 200 | 1,000 |
| 2D CDR interpolation | Ours | 21,828 | 3,000 | 15,000 |
| | DeepONet | 24,828 | 9,828 | 15,000 |
| 2D CDR extrapolation | Ours | 21,828 | 1,024 | 3,072 |
| | A-FNO | 22,264 | 1,024 | 3,072 |
| | F-FNO | 22,264 | 1,024 | 3,072 |
| | FNO | 22,264 | 1,024 | 3,072 |
| | DeepONet | 22,264 | 1,024 | 3,072 |
| 2D biharmonic interpolation | Ours | 19,348 | 3,000 | 15,000 |
| | DeepONet | 22,348 | 9,828 | 15,000 |
| Burgers' extrapolation | Ours | 2,508 | 256 | 256 |
| | FNO | 2,560 | 256 | 256 |
| | DeepONet | 2,560 | 256 | 256 |

The hyperparameters for our model and the baselines in the 1D CDR experiment can be found in Table 19. Among them, the hyperparameters used in the reduced version setting of Hyper-LR-PINN, as shown in Table 2, are marked separately with an (*). Table 20 presents the hyperparameters of baselines and our model used in the 2D CDR experiments. It indicates the best hyperparameter settings for each task and prior used separately. Table 22 presents the best hyperparameters settings of our model and DeepONet used in the 2D biharmonic experiments.

Table 19: Hyperparameter used in 1D CDR experiments.

| Model | Hyperparameter Name | Best Hyperparameter Setting |
|---|---|---|
| Ours | Transformer layers number | 3 |
| | Transformer hidden size | 32 |
| Hyper-LR-PINN | phase 1 layers number | 2 |
| | phase 2 layers number | 2 |
| | hidden dimension | 50 (*20) |
| P$^2$INN | parameter encoder layers number | 4 |
| | spatiotemporal coordinate encoder layers number | 3 |
| | decoder layers | 7 |
| PINN-prior | training loss threshold | $1 \times 10^{-3}$ |
| | maximum training epoch | 100 |

Table 20: Hyperparameters used in 2D CDR experiments.

| Model | Hyperparameter Name | Best hyperparameter setting in interpolation task | | Best hyperparameter setting in extrapolation task | |
|---|---|---|---|---|---|
| | | Numerical prior | PINN prior | Numerical prior | PINN prior |
| Ours | Transformer layers number | 3 | 3 | 3 | 5 |
| | Transformer hidden size | 64 | 64 | 64 | 64 |
| DeepONet | branch net depth | 3 | 4 | 4 | 4 |
| | trunk net depth | 5 | 5 | 4 | 5 |
| | hidden size | 64 | 128 | 64 | 64 |
| FNO | layers number | - | - | 4 | 4 |
| | hidden size | - | - | 64 | 64 |
| F-FNO | layers number | - | - | 20 | 20 |
| | hidden size | - | - | 256 | 256 |
| A-FNO | layers number | - | - | 8 | 16 |
| | hidden size | - | - | 64 | 64 |
| PINN prior | training loss threshold | $1 \times 10^{-4}$ | | | |
| | maximum training epoch | 200 | | | |

Table 21: Hyperparameter used in 2D biharmonic equation experiments.

| Model | Hyperparameter Name | Best Hyperparameter Setting |
|---|---|---|
| Ours | Transformer layers number | 3 |
| | Transformer hidden size | 64 |
| DeepONet | branch net depth | 4 |
| | trunk net depth | 5 |
| | hidden size | 128 |

Table 22: Hyperparameter used in 2D Burgers' equation experiments.

| Model | Hyperparameter Name | Best Hyperparameter Setting |
|---|---|---|
| Ours with PINN loss | Transformer layers number | 7 |
| | Transformer hidden size | 512 |
| Ours | Transformer layers number | 3 |
| | hidden size | 1,024 |
| FNO | layers number | 6 |
| | hidden size | 256 |
| DeepONet | branch net depth | 5 |
| | trunk net depth | 5 |
| | hidden size | 512 |

# O   MODEL ARCHITECTURE

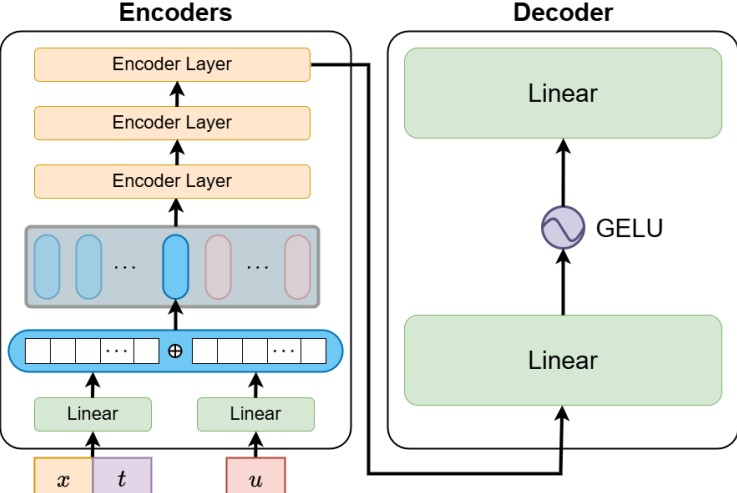

Figure 7: A schematic diagram of the Transformer architecture. The detailed model architecture, comprising an encoder and decoder, is presented as an extension of Figure 2. The figure shows the use of three encoder layers, but the number of encoder layers was determined through hyperparameter optimization.

The terminology used in this description corresponds to the terms used in main text and carries the same meaning. Additionally, we assume the training scenario and refers to the data as $D$ and $T$. The model is fundamentally designed based on the *vanilla* Transformer architecture (Vaswani, 2017). Initially, the spatiotemporal coordinates and solution points of the input $D \cup T$ are separately embedded using linear layers. These embeddings are then concatenated and provided as input to the encoder layer. In each encoder layer, which is basically a Transformer block, the source mask is utilized to enable both self-attention and cross-attention mechanisms which are shown with red and blue arrows in Figure 2. The mask is used to enable the self-attention among $D$ while allowing the cross-attention from $T$ to $D$. The number of the blue rods for $D$ and the red rods for $T$ in this figure is determined by a fixed ratio during training as described in D. Furthermore, we omit the use of positional encoding to maintain equivariance between data points, since they are derived from the same PDE prior. Instead of multi-head attention, the encoder employs a single-head attention with several stacked layers. While the figure illustrates the use of three encoder layers, the number of encoder layers was optimized as a hyperparameter. The decoder adopts a multi-layer perceptron (MLP) structure to produce the inferred solutions effectively.

