# OpenReview forum: "MaD-Scientist: AI-based Scientist solving Convection-Diffusion-Reaction Equations Using Massive PINN-Based Prior Data"
_ICLR.cc/2025/Conference — Submitted to ICLR 2025_

### Official Review · Reviewer_QAiw · 2024-10-30

**Soundness:** 3
**Presentation:** 3
**Contribution:** 2
**Rating:** 5
**Confidence:** 2

**Summary:**

This paper introduces an in-context learning architecture for scientific foundation models. The proposed method incorporates Bayesian inference into the prediction by pre-training on PINN-based low-cost and noisy approximated data, which is referred to as the PINN-prior. The PINN is initialized as a general PDE with derivatives up to a certain degree. Initializing the PINN with a general PDE helps to overcome the fact that the underlying governing equations of physical systems are often unknown; instead, only the observations of the underlying systems are available. At the base, the method uses a transformer architecture, where a self-attention mechanism is used to capture relationships among known data points, and a cross-attention mechanism is used to extrapolate and infer solutions for unseen points without fine-tuning.

**Strengths:**

1. This paper integrates several non-trivial concepts of the scientific foundation model, approximated prior data, Bayesian inference, and pre-training that are already well-established in scientific machine learning.
2. The concept of using PDE dictionaries in the PINN-prior for pre-training is original.
3. This paper tries to address the challenges of the inherent noisiness of the data used for training.

**Weaknesses:**

According to me, some parts of the paper need further clarification. Please see the comments below:
1. The underlying concept of using approximated prior data is a concept that has been introduced previously in scientific machine learning. This concept already exists in multi-fidelity learning, where a low-fidelity solver is first trained on a large number of available low-fidelity data, and the approximate predictions of low-fidelity solver are used in conjunction with the small high-fidelity data to train a fast high-fidelity emulator [1,2]. Except for the difference between the purely data-driven nature of the multi-fidelity learning methods and the partial physics-constrained nature of the proposed framework, I do not see any significant improvement.  Further clarification is required to understand the novelty of the proposed method.
2. The authors mention the "scientific foundation model" at several places in the paper. The purpose of the foundation models is to pre-train on a large number of datasets, which in this case is analogous to multiple PDEs, and then efficiently fine-tune or directly predict on unseen PDEs without sacrificing prediction accuracy. However, the authors consider only a simple CDR equation for pre-training and special cases of the CDR equation like convection, diffusion, and reaction equations for testing. This raises concerns about whether the methods can genuinely aid in foundation modeling, highlighting the need for more rigorous case studies to properly assess the effectiveness of the proposed PINN-prior framework. To be able to clearly appreciate the method's potential as a foundation model, in addition to the CDR equation, it is necessary to simultaneously pre-train on a few of more complex PDEs like the Kuramoto–Sivashinsky equation, Schrodinger equation, Korteweg–De Vries equation, Navier-Stokes equation, Burgers' equation, etc. and test on remaining PDEs.
3. In the PINN dictionary, the authors consider the exact derivative terms of the CDR equation. This is unlikely to be available in a practical scenario, which is also mentioned by the authors in the motivation. Thus, the performance of the PRIOR in the presence of a large number of dictionary functions needs to be inspected to understand further the potential of the proposed framework for scientific foundation modeling. The authors need to conduct additional experiments with a larger and more diverse set of dictionary functions to better simulate real-world scenarios where exact terms may not be known.
4. Many scientific foundation models have already been developed for pre-training and testing on multiple physical systems [3-6]. However, the authors do not mention them in this paper. Given the existing literature on foundation models, the authors should consider comparisons with a few of the existing scientific foundation models. This will further improve the contents of the paper.

[1] Meng, Xuhui, and George Em Karniadakis. "A composite neural network that learns from multi-fidelity data: Application to function approximation and inverse PDE problems." Journal of Computational Physics 401 (2020): 109020.\
[2] Howard, Amanda A., et al. "Multifidelity deep operator networks for data-driven and physics-informed problems." Journal of Computational Physics 493 (2023): 112462.\
[3] McCabe, Michael, et al. "Multiple physics pretraining for physical surrogate models." arXiv preprint arXiv:2310.02994 (2023).\
[4] Tripura, Tapas, and Souvik Chakraborty. "A foundational neural operator that continuously learns without forgetting." arXiv preprint arXiv:2310.18885 (2023).\
[5] Hao, Zhongkai, et al. "Dpot: Auto-regressive denoising operator transformer for large-scale pde pre-training." arXiv preprint arXiv:2403.03542 (2024).\
[6] Herde, Maximilian, et al. "Poseidon: Efficient Foundation Models for PDEs." arXiv preprint arXiv:2405.19101 (2024).

**Questions:**

Please see below questions on the paper content:
1. line 078. The authors claim that their goal is "to predict solutions from observed quantities, such as velocity and pressure, without relying on governing equations". However, any data-driven ML model learns solutions from observed quantities without relying on governing physics. Therefore, why use a PINN with approximate PDE dictionaries while robust methods like neural operators are already there?
2. line 269. In Eq. (12), define $\tilde{u}(x_T^{(j)},t_T^{(j)})$. Is it the same as $\tilde{u}(\alpha)$, or does it denote the observed solution field?
3. line 271. "...our model only needs observed quantities". The correct derivatives, as in the original CDR equation, are considered in Eq. (8). Thus, this sentence seems to be vague to me. Consider a large number of derivatives and their polynomials in Eq. (8) and then generate prior data to see the performance of the proposed method.
4. line 302. it is written $D$ requires the solution u. Is $u$ the true observed quantity or the PINN-approximated data?
5. line 323. What value of $m$ is taken in the examples?
6. line 372. Is it supposed to be section 4.2.1?
7. Table 3. For 100% PINN-prior, the error in the convection equation is as high as 16%. This raises questions about the framework's ability to generalize effectively over a large number of unseen systems.
8. Is there any reason similar to the above why the "TIME DOMAIN EXTRAPOLATION" study is not carried out on diffusion and reaction equations?
9. Fig. 3. This is a continuation of comment 7. See that the time extrapolation results for the convection equation is more than 50%, which again raises questions about the framework's generalization ability.
10. Fig. 4. The solution field on the right is too smooth to draw a conclusion about the extrapolation ability.
11. line 480. Why is the $\beta$ = $\nu$ = 0 taken? I think a non-zero value will make the problem more challenging.
12. line 483. It is not fair to say "it can generalize to unseen PDEs" since the model is pre-trained on the CDR PDE, which is nothing but a mixture of convection, diffusion, and reaction.
13. line 487. How many training samples are taken?
14. Table 4. A central goal of the paper is to achieve zero-shot inference for predicting PDE solutions. However, when the model is pre-trained on a mixture of reactions but tested on different reactions separately, the error increases significantly beyond 6% in most of the cases. This indicates that the proposed framework may also need fine-tuning or few-shot learning.

---

> ### Author Response · Authors · 2024-11-21
> **Response to reviewer QAiw**
>
> Thank you for the valuable feedback on our paper. We hope that our responses will address the reviewer’s questions effectively.
>
> **[W1]** As you mentioned, the referenced papers do utilize low-fidelity data. However, their approaches and objectives differ from those of our model. We have compared these aspects in the table below.
>
> | Aspects                              |  Ours  |  [1]  |  [2]  |
> | ------------------------------------ |:------:|:-----:|:-----:|
> | Use high-fidelity data on training   |   X    |   O   |   O   |
> | Use govern equation on training      |   X    |   O   |   O   |
> | Solve multiple PDEs                  |   O    |   X   |   X   |
> | Solve PDEs with unseen parameters    |   O    |   X   |   X   |
>
> First, we compare in terms of whether high-fidelity data is used during the training phase. Our model can be properly trained without any high-fidelity data during training. On the other hand, the other two models use high-fidelity data during training.
> Second, we compare in terms of whether information about the governing equation is utilized. In our model, no information about the governing equation is used at any stage. Our model learns solely based on observed data. However, the other two models use the PINN loss during training.
> Third, we compare in terms of whether the model solves multiple PDEs. Since our model is a foundation model, it is trained with priors generated from multiple PDEs. On the other hand, the other two models are trained with data from only one target PDE.
> Lastly, we compare in terms of whether the model solves PDEs with unseen parameters. Our model can generalize to PDEs with unseen parameters. However, the other two models are tested only on the trained PDE.
>
> [1] Meng, Xuhui, and George Em Karniadakis. "A composite neural network that learns from multi-fidelity data: Application to function approximation and inverse PDE problems." Journal of Computational Physics 401 (2020): 109020.
>
> [2] Howard, Amanda A., et al. "Multifidelity deep operator networks for data-driven and physics-informed problems." Journal of Computational Physics 493 (2023): 112462.
>
>
> **[W2]** As you mentioned, since our original paper only presented experiments on 1D CDR, it was necessary to explore the broader potential of our model. In response to this necessity, we extend the dimension to conduct experiments on 2D CDR. Moreover, we conduct experiments on the biharmonic equation and Burgers' equation. In these experiments, our model demonstrated relatively strong performance compared to the baselines. Details for the experiments can be found in Appendices I, J, and K of the revised paper. While we have not pre-trained the model on a variety of complex PDEs as suggested, these additional experiments confirm that our model is not limited to working effectively on only 1D CDR.
>
>
> **[W3]** We used the CDR equation to simplify the experimental setup and for comparison with existing methods such as Hyper-LR-PINN and P$^2$INN. However, our model did not receive any information about the equation during training. It was trained solely on observed data points sampled from the prior, and this reflects real-world scenarios.
>
> In the 1D CDR experiments, we conducted tests with coefficients $\beta, \nu, \rho \in [1, 20] \cap \mathbb{Z}$ , which resulted in a dictionary containing 8,000 PDEs. For the 2D CDR experiments in Appendix I of the revised paper, we use the coefficients with $\beta_x, \beta_y, \nu_x, \nu_y, \rho \in [1, 3] \cap \mathbb{Z}$, which formed a dictionary of 243 PDEs. In both cases, we believe that we have demonstrated the potential of our model as a scientific foundation model by using sufficiently large dictionaries.
>
> Additionally, we conduct experiments not only on CDR but also on Burgers' equation and the biharmonic equation, proving that our model is not limited to CDR applications. The details of the experiments are in Appendix J and K in the revised paper.

---

> > ### Author Response · Authors · 2024-11-21
> >
> > **[W4]** As you mentioned, the referenced papers are scientific foundation models. However, they differ significantly from our model in various aspects. We compare them using four criteria in the table below:
> >
> > | Aspect                       | Ours | MPP[1] | NCWNO[2] | Dpot[3] | Poseidon[4] |
> > | ---------------------------- | :--: | :----: | :------: | :-----: | :---------: |
> > | Zero-shot generalization     |  O   |   X    |    O     |    X    |    O        |
> > | Utilizes Bayesian inference  |  O   |   X    |    X     |    X    |    X        |
> > | Noisy prior handling         |  O   |   X    |    X     |    X    |    O        |
> > | Utilizes low-cost prior      |  O   |   X    |    X     |    X    |    X        |
> >
> > First, we compare in terms of whether the model leverages zero-shot generalization. Our model utilizes zero-shot generalization, whereas [1, 3] do not rely on zero-shot capabilities.
> > Second, we compare in terms of whether the model is based on Bayesian inference. Our model is based on Bayesian inference, but other models are not.
> > Third, we compare in terms of whether the model can handle noisy prior data. Our model has been experimentally shown to maintain performance even when trained with noisy data. However, [1-3] have not demonstrated robustness in this regard.
> > Lastly, we compare in terms of whether the model uses low-cost prior data. Our model showed that it can maintain a certain level of performance even when trained with low-cost priors, such as PINN priors, instead of hard-to-obtain numerical data. In contrast, the other models have not explored or analyzed this possibility.
> >
> > [1] McCabe, Michael, et al. "Multiple physics pretraining for physical surrogate models." arXiv preprint arXiv:2310.02994 (2023).
> >
> > [2] Tripura, Tapas, and Souvik Chakraborty. "A foundational neural operator that continuously learns without forgetting." arXiv preprint arXiv:2310.18885 (2023).
> >
> > [3] Hao, Zhongkai, et al. "Dpot: Auto-regressive denoising operator transformer for large-scale pde pre-training." arXiv preprint arXiv:2403.03542 (2024).
> >
> > [4] Herde, Maximilian, et al. "Poseidon: Efficient Foundation Models for PDEs." arXiv preprint arXiv:2405.19101 (2024).
> >
> > **[Q1]** We recognize that data-driven methods, such as FNO and DeepONet, are effective. However, we chose to use PINN-based approximate PDE dictionaries because they align closely with the goals of our approach. Our objective is to develop a model that can learn from low-cost, physics-based approximate data, which makes PINNs an ideal choice in scenarios where high-fidelity data or full numerical solutions may be cost-prohibitive or inaccessible [1-2]. The PINN framework allows us to incorporate approximate, physics-guided data without excessive costs, providing a flexible basis for scientific applications.
> > Furthermore, we conduct comparative experiments with neural operators, including FNO and DeepONet, and find that our PINN-based approach demonstrates better performance. This makes PINN a more suitable choice for building a scalable and adaptable foundation model.
> >
> > [1]Bolandi, Hamed, et al. "Physics informed neural network for dynamic stress prediction." Applied Intelligence 53.22 : 26313-26328 (2023).
> >
> > [2]Weikun, D. E. N. G., et al. "Physics-informed machine learning in prognostics and health management: State of the art and challenges." Applied Mathematical Modelling 124: 325-352 (2023).
> >
> > **[Q2]** It is the same as $\tilde{u}(\alpha)$.

---

> > > ### Author Response · Authors · 2024-11-21
> > >
> > > **[Q3]** We acknowledge the initial vagueness of the sentence, and it has been revised in the paper. Our model, trained with extensive prior data, is expected to infer unknown solution points based on given solution points alone. In this context, the phrase ‘...our model only needs observed quantities’ means that the model can infer unknown solutions using observed data only, without the need of additional information on the PDE equation, e.g., Hyper-LR-PINN and P$^2$INN require the parameter information of the target parameterized PDE to predict. Therefore, one can consider that our model is suitable for spatiotemporal interpolation/extrapolation tasks with no additional information except observed data. Since our model is trained with a vast set of prior PDEs, it can quickly understand the underlying dynamics of the observations (or the context) in $\widetilde{D}$ and predict for a test query.
> > >
> > > Furthermore, our model does not take derivatives or other derived variables as inputs. When we refer to "observed quantities," we mean coordinates and the quantities, e.g., velocity, pressure, and so on, at those coordinates. The model is designed to learn based on these observations and can derive appropriate results without additional derivative information, e.g., symbolic PDE equation.
> > >
> > > To evaluate our model's performance on PDEs with higher-order derivative terms, we conduct an interpolation task experiment on the 2D biharmonic equation, which includes fourth-order derivative terms. In this experiment, we compare the performance of our model with DeepONet, and our model demonstrates superior performance. This result confirms that our model's in-context learning is also effective for PDEs with higher-order derivative terms. For experiment details, please refer to Appendix J of the revised paper.
> > >
> > > **[Q4]** The data is essentially PINN-approximated but varies across sections. In Section 4.2.1, we used the exact data to establish our model’s performance. Then, in Section 4.2.2, we mixed the PINN-approximated data with the exact data in ratios of 0%, 20%, 40%, 60%, 80%, and 100% to demonstrate the effectiveness of using the PINN-approximated data. In the subsequent sections, we relied solely on the PINN-approximated prior data.
> > >
> > > **[Q5]** $m$ indicates the number of nonzero coefficients in each type of equation. For instance, $m=1$ for the convection, diffusion, and reaction equation, respectively, $m=2$ for the convection-diffusion and reaction-diffusion equation, and $m=3$ for the convection-diffusion-reaction equation in Table 2.
> > >
> > > **[Q6]** Section 4.2.1. is correct. we have fixed it in the revised paper.
> > >
> > > **[Q7, Q9]** In our paper, the convection equation solution is the most complex and distinct compared to other CDR equations' solutions, which is the reason for the lower inference accuracy. This is also the case in other works, such as [1] and [2] (papers), and [3] (a book). Training all equation types with an equal weight, without fine-tuning, limits the model’s ability to generalize to the convection equation. However, by expanding the prior data and its PDE solution space, the model's capacity to infer solutions for the convection equation improves.
> > >
> > > [1] Krishnapriyan, A.S., et al. Characterizing possible failure modes in physics-informed neural networks. Neural Information Processing Systems (2021).
> > >
> > > [2] Morton, K. (n.d.). *Numerical Solution Of Convection-Diffusion Problems*. Chapman and Hall/CRC.
> > >
> > > [3] Douglas, J., Jr, et al. Numerical Methods for Convection-Dominated Diffusion Problems Based on Combining the Method of Characteristics with Finite Element or Finite Difference Procedures. SIAM Journal on Numerical Analysis, 19(5), 871–885 (1982).
> > >
> > > **[Q8]** In the time domain extrapolation study, the model performs extrapolation over the interval $0.6 \le t \le 1.0$. However, the solution for the 1D diffusion or reaction equation with an initial condition of $1+\sin(x)$ is approximately $1$ at most points in the spatial domain.

---

> ### Author Response · Authors · 2024-11-21
>
> **[Q10]** Although the extrapolation conducted in this task was on a smooth domain, we found it meaningful as our model outperformed other baselines and also showed superior performance compared to neural operator baselines in the additional 2D CDR experiment. More challenging experiments, such as boundary layer problems, are important topics in their own right and will be addressed in future work.
>
> Additionally, to evaluate our model's performance on a PDE exhibiting singular behavior in its solution profile, we conduct an extrapolation task experiment on the Burgers equation with shock formation. In this experiment, we compare the performance of our model with FNO and DeepONet. Our model outperforms DeepONet but falls short of FNO. This indicates that our model's cross-attention structure may be insufficient to infer unexpected patterns.
>
> To address this limitation, we adopt the approach suggested by Reviewer FgQz, incorporating the PINN loss into our model during training. As a result, the enhanced model outperforms the original model across all metrics and surpasses FNO in two metrics. This demonstrates the potential of PINN loss as a promising direction for improving our model in future work. Detailed information about this experiment has been added in Appendix K of the revised paper.
>
> **[Q11]** Before dealing with more challenging problems, it is essential to verify that our model can accurately predict the linear combination of various reaction terms. To this end, we expand our experiments to explore nonzero values for both convection and diffusion terms. In other words, we conduct additional experiments on 1D CDR with diverse reaction terms using fine-tuning. We identify that while our model performs well with only pre-training, fine-tuning can further improve its performance when higher accuracy is desired. Detailed information about this experiment has been added in Appendix L.2 of the revised paper.
>
> **[Q12]** Apologies for the confusion. What we intended was the generalization within the CDR framework to unseen parameters, rather than to completely different types of unseen PDEs. We have fixed this sentence in the revised paper based on your feedback.
>
> **[Q13]** Apologies for not including experimental details in the main text of the paper. As mentioned in the general response, we have added details of the experiments in Appendix N of the revised paper. During the training phase, we sampled 800 collocation points, 256 initial condition points, and 200 boundary condition points to train our model. In the test phase, we used 200 collocation points as observed data to infer solutions for 1,000 non-overlapping points where the solution is unknown.
>
> **[Q14]** You are absolutely correct that fine-tuning or few-shot learning can further improve model performance. However, in this paper, especially in Section 4.3, we aimed to emphasize the model’s capability for in-context learning when trained with a linear combination of different reaction terms. This approach is a novel yet essential attempt to expand the parameter space and demonstrate the model’s potential to perform inference effectively even in a zero-shot setting. We wanted to highlight this possibility, which we believe is a valuable step towards more flexible and generalizable PDE solution inference.
>
> The 6% error mentioned is due to the tested parameter space. In the last three columns of Table 4, the PDE parameters tested were not seen during training, while the first column includes parameters seen in training. Additionally, the range of the tested PDE parameters lies outside the range of the training parameters, making it an extrapolation of the parameter space. Therefore, we believe this error does not directly critique our model’s in-context learning capability. To address this, we have added an additional experiment with fine-tuning as a complement in the revised paper. Detailed information about this experiment has been added in Appendix L.1 of the revised paper.

---

> > ### Author Response · Authors · 2024-11-27
> >
> > Dear Reviewer QAiw,
> >
> > Thank you for your valuable feedback, which has greatly contributed to improving our paper. Your thoughtful questions allowed us to consider additional aspects necessary for our paper. In particular, the comparisons with the papers you suggested helped us further emphasize the strengths of our work. Additionally, the supplementary experiments provided another opportunity to confirm the potential of our model.
> >
> > We gently remind you to review our responses, as submition deadline of the revised paper is November 27. We have worked hard to address your feedback in detail. We would greatly appreciate it if you could let us know whether our responses have sufficiently resolved your concerns and consider updating your score, or inform us if you have any further questions or concerns.
> >
> > Best regards,
> > The Authors

---

> ### Comment · Area_Chair_NeY9 · 2024-11-27
>
> Dear Reviewer,
>
> The authors have provided their rebuttal to your comments/questions. Given that we are not far from the end of author-reviewer discussions, it will be very helpful if you can take a look at their rebuttal and provide any further comments. Even if you do not have further comments, please also confirm that you have read the rebuttal. Thanks!
>
> Best wishes,
> AC

---

> > ### Comment · Reviewer_QAiw · 2024-11-27
> >
> > Thank the author for the response and effort on the additional tasks and comparison with neural operators. I have increased my score accordingly. Thank you.

---

> > > ### Author Response · Authors · 2024-11-28
> > > **Thank you for the positive feedback**
> > >
> > > Dear Reviewer QAiw,
> > >
> > > We are pleased to hear that all your concerns have been thoroughly addressed. Your valuable feedback has significantly contributed to enhancing the quality of our work. We greatly appreciate your involvement in the discussion and your recognition of the role it played in refining our paper.
> > >
> > > Although you adjusted the rating based on our response, it is still below the acceptance threshold. If you have any further questions or comments, please don’t hesitate to reach out.
> > >
> > > Best regards,
> > > The Authors

---

### Official Review · Reviewer_WwXL · 2024-10-31

**Soundness:** 2
**Presentation:** 2
**Contribution:** 3
**Rating:** 5
**Confidence:** 3

**Summary:**

The paper presents a scientific foundation model to solve convection-diffusion-reaction equations in a zero-shot setting. Its main novelty is that the PDE solution data used to train the model is itself generated by a PINN solver, and is therefore subject to noise and measurement error. The authors demonstrate the the model can outperform baselines and make accurate predictions under parameter distribution shifts and time-domain extrapolation.

**Strengths:**

1. This work validates the hypothesis that foundation models are able to generalize even under noisy prior data, which has been thoroughly demonstrated in NLP but has been largely unexplored in scientific ML.

2. The model performs well even under substantial distribution shifts, which suggests that it is really learning to learn as opposed to memorizing the distribution of the training data.

**Weaknesses:**

1. The authors provide very few details on their architecture, other than that they use a transformer with self-attention and cross-attention. This makes it very difficult to meaningfully compare their model with other baselines. I would suggest adding a detailed description of the architecture to the appendix.

2. The class of PDEs considered is quite restrictive, since it is a family of 1D equations parameterized by relatively few variables. It doesn't seem too surprising that a sufficiently complex foundation model can learn this family of equations.

Typos: In Theorem 2.1, I think the first inequality should say that $\mathbb{E}[KL(p_{\widehat{\theta}}(\cdot | x, D_n) \parallel \pi(\cdot | x, D_n))] < \epsilon$ rather than $\widehat{\theta} < \epsilon$.

**Questions:**

1. Can you clarify specifically what you mean by 'zero-shot inference'? In Figure 2 (Inference Phase) it looks like you prompt your model with pairs $(x_i,t_i,u_i)$, and thus require data from the operator to be inferred in order to make predictions.

2. Can you explain your rationale for using the L^2 norm as an evaluation metric, as opposed to the H^1 norm or a more PDE-specific metric?

---

> ### Author Response · Authors · 2024-11-21
> **Response to reviewer WwXL**
>
> Thank you for the valuable feedback on our paper. We hope that our responses will address the reviewer’s questions effectively.
>
> **[W1]** Please refer to the general response. We have added description of our model and a simplified figure illustrating the detailed model structure in Appendix O of the revised paper. We also have revised Figure 2 in the revised paper.
>
> **[W2]** Please refer to the general response. We have incorporated an experiment involving the biharmonic equation with higher-order derivative terms and the Burgers equation with solution profile that exhibit singular behavior. These additions demonstrate our model's capablility for in-context learning across a diverse class of PDEs.
>
>
> **[Q1]** The concept of "zero-shot inference" in our paper refers to inferring solutions for a partial differential equation (PDE) without additional information except the target PDE itself. Unlike the in-context operator learning (ICON) introduced by L. Yang et al. (2023), our model does not rely on "demos" (few-shot examples with various initial/boundary conditions regarding the target operator to learn) during inference. ICON learns using diverse conditions and quantity-of-interest (QoI) pairs, i.e., demos, for a hidden PDE and employs these demos to predict answers for a new test condition, which is a few-shot learning.
>
> In contrast, our model infers solutions to unseen PDEs solely based on the prior knowledge learned from a set of prior PDEs. This prior knowledge encapsulates generalizable patterns from the solution space of PDEs, enabling zero-shot inference without additional demos. The prompts shown in Figure 2, which consist of the information about observed solution points, represent the only input provided for the unseen PDE. Importantly, these prompts are not analogous to few-shot demos as used in ICON. This distinction underscores the zero-shot capability of our approach, which eliminates the need for PDE-specific training or demo-based fine-tuning.
>
> **[Q2]** In machine learning, the $L_1$ absolute error, $L_2$ absolute error and $L_2$ relative error are commonly used as a base metric. For example, [1-2] used the $L_2$ relative error, [3] employed the $L_1$ absolute error, and [4] utilized the $L_1$ absolute error, MSE, and $L_2$ relative error. Thus we have used the $L_1$ absolute error and $L_2$ relative error. Also, we used $L_2$ relative error for performance comparison with methodologies presented in other papers, as this metric enables a direct comparison.
> Instead of the $H_1$ norm, we have additionally provided the $L_{\infty}$ relative error in the revised paper.
>
> [1] Kang, Namgyu, et al. "Pixel: Physics-informed cell representations for fast and accurate pde solvers." Proceedings of the AAAI Conference on Artificial Intelligence. Vol. 37. No. 7. 2023.
>
> [2] Lee, Jae Yong, Sung Woong Cho, and Hyung Ju Hwang. "HyperDeepONet: learning operator with complex target function space using the limited resources via hypernetwork." arXiv preprint arXiv:2312.15949 (2023).
>
> [3] Herde, Maximilian, et al. "Poseidon: Efficient Foundation Models for PDEs." arXiv preprint arXiv:2405.19101 (2024).
>
> [4] Yang, Liu, et al. "In-context operator learning with data prompts for differential equation problems." Proceedings of the National Academy of Sciences 120.39 (2023): e2310142120.

---

> > ### Author Response · Authors · 2024-11-27
> >
> > Dear Reviewer WwXL,
> >
> > Thank you for your valuable feedback, which has greatly contributed to enhancing our paper. Based on your comments, we have revised Figure 2 and corrected typo. Also, we have included a detailed description of our model's structure in the revised version of the paper.
> >
> > This is a gentle reminder, as the deadline for submitting the revised paper is November 27. We made significant efforts to address your inquiries in detail. We would greatly appreciate it if you could let us know whether our responses have sufficiently addressed your concerns and consider reconsidering your evaluation, or inform us if you have any further questions.
> >
> > Best regards,
> > The Authors

---

> > ### Comment · Reviewer_WwXL · 2024-11-27
> > **Reply to the authors**
> >
> > I thank the authors for addressing my questions. The use of the phrase 'zero-shot inference' has been made clear to me, and while I still feel that the H^1 norm is still a relevant metric for the PDE problems considered in the paper, I respect the choice of the authors to use the more standard L^1 and L^2 norms. My overall assessment of the paper remains the same and I will maintain my score.

---

> ### Comment · Area_Chair_NeY9 · 2024-11-27
>
> Dear Reviewer,
>
> The authors have provided their rebuttal to your comments/questions. Given that we are not far from the end of author-reviewer discussions, it will be very helpful if you can take a look at their rebuttal and provide any further comments. Even if you do not have further comments, please also confirm that you have read the rebuttal. Thanks!
>
> Best wishes,
> AC

---

> ### Author Response · Authors · 2024-11-27
> **Additional response to reviewer WwXL**
>
> Dear Reviewer WwXL,
>
> We are glad that your questions regarding zero-shot inference have been resolved. We also appreciate your understanding of the metrics we have used. Nevertheless, to address your point about the necessity of the H1 norm, we have attempted to evaluate the error using the H1 norm during the rebuttal period for the 2D CDR interpolation task. For this evaluation, we use the settings that had previously shown the best performance for each model and evaluate on a new test dataset in the form of grid data.
> The results have been incorporated into Appendix I.2 and I.4 of the revised paper. The results demonstrate that our model outperforms the baselines with respect to the H1 norm as well.
>
> We are currently working on evaluating other experiments using the H1 norm. We will do our best to present these results before the discussion period concludes.

---

> ### Author Response · Authors · 2024-12-03
> **Additional response to reviewer WwXL**
>
> Dear Reviewer WwXL,
>
> As mentioned in our previous additional response, we have evaluated the $H_1$ norm in another experiment. This experiment was conducted on the time domain extrapolation task with unseen parameters in 1D convection equation. The results are as follows:
>
> |$\beta$|   Ours   | Hyper-LR-PINN | P$^2$INN |
> |-------|:----------:|:---------------:|:----------:|
> |  1.5  |  0.4064  |    0.8291     |  0.8404  |
> |  2.5  |  0.8180  |    0.6279     |  0.8000  |
> |  3.5  |  0.2751  |    0.7165     |  0.8259  |
> |  4.5  |  0.3719  |    0.8536     |  0.8699  |
> |  5.5  |  0.4098  |    0.9797     |  0.8115  |
> |  6.5  |  0.3120  |    1.0622     |  1.3933  |
> |  7.5  |  0.6129  |    0.3217     |  1.3819  |
> |  8.5  |  0.6875  |    1.2644     |  1.3703  |
> |  9.5  |  0.5261  |    1.4140     |  1.4659  |
> | 10.5  |  0.8188  |    1.5769     |  1.6317  |
> | Average| 0.5239  |    0.9646     |  1.1391  |
>
> Since this experiment was conducted on equations with unseen parameters, the absolute values are slightly higher. However,  our model still demonstrates superior performance compared to the baselines. These results will be included in the paper in the future, as the deadline for revising the current paper has already passed.
>
> The discussion period end on December 2(AoE). If you have any additional questions, please let us know before the deadline. We have continuously worked throughout the discussion period to address your concerns and questions.
>
> Best regards,
> The Authors

---

### Official Review · Reviewer_sZQg · 2024-11-04

**Soundness:** 3
**Presentation:** 3
**Contribution:** 3
**Rating:** 8
**Confidence:** 2

**Summary:**

This paper presents a scientific foundation model to solve PDEs with scarce data by training on low-cost data. This methodology is proven to be amenable to zero-shot learning with high performance, as evidenced by benchmark experiments across various kinds of PDEs.

**Strengths:**

- The paper is well organized and written.
- Encouraging results across datasets.
- To have a genearlizable zero-shot PDE solver trained from low-cost data would have high impact on the physical modeling community.

**Weaknesses:**

- Figure 2. used a lot of page real estate to demonstrate concepts not unique to the paper. I would perhaps highly the innovation of this paper here.
- I feel like to brand this as a _scientific_ foundation model is slightly overselling it. To solve PDEs is hardly equal to _science_.

**Questions:**

- I feel like the idea behind Theorem 2.1 is rather intuitive. And it is a little out-of-place here. I don't see it referenced at all in the text. Could you motivate this theorem a bit more?

---

> ### Author Response · Authors · 2024-11-21
> **Response to reviewer sZQg**
>
> Thanks for your valuable feedback. We have added experiments related to the theorem, providing not only a mathematical proof but also empirical evidence.
>
> **[W1]** As mentioned in the general response, we have added the description of our model and a simplified figure in Appendix O of the revised paper. We have also revised Figure 2 in the revised paper.
>
>
> **[W2]** As we mentioned in the introduction, various models are already categorized as Scientific Foundation Models (SFMs). However, each of these models covers only specific tasks in the realm of science. For example, [1, 2] focus solely on tasks related to the medical domain, and [3] is dedicated to audio-related tasks. Similarly, solving PDEs is a subset of science, and it is a highly studied area. Consequently, models like [4, 5] are referred to as SFMs for solving PDEs. Therefore, as our model also belongs to this category of foundation models for solving PDEs, we described it as an SFM. However, we will make it clear that our model is a scientific founcation model for solving PDEs.
>
> [1] Xie, Qianqian, et al. "Me llama: Foundation large language models for medical applications." arXiv preprint arXiv:2402.12749 (2024).
>
> [2] Moor, Michael, et al. "Foundation models for generalist medical artificial intelligence." Nature 616.7956 (2023): 259-265.
>
> [3] Yang, Dongchao, et al. "Uniaudio: An audio foundation model toward universal audio generation." arXiv preprint arXiv:2310.00704 (2023).
>
> [4] Yang, Liu, et al. "In-context operator learning with data prompts for differential equation problems." Proceedings of the National Academy of Sciences 120.39 (2023): e2310142120.
>
> [5] Herde, Maximilian, et al. "Poseidon: Efficient Foundation Models for PDEs." arXiv preprint arXiv:2405.19101 (2024).
>
> **[Q1]** As mentioned in the general response, we have added an experiment on the model's sensitivity to the number of the given data $\widetilde{D}$ during testing in Appendix M of the revised paper.
> This result demonstrates the sensitivity of the neural network's posterior distribution approximation to the data size ($D_n$). As the data size increases, the network's approximation to the posterior distribution becomes quickly accurate, converging to the expected value under the prior distribution. This sensitivity to the data reflects the consistency and robustness of the Bayesian inference process.
>
> Our model leverages this observation by performing Bayesian inference that incorporates prior data, allowing it to infer spatiotemporal points that align with an appropriate partial differential equation (PDE) solution under given conditions. Experimental results in Appendix M confirm this behavior, showing how, as ($D_n$) increases, the network’s solution converges to the true underlying solution.

---

> > ### Author Response · Authors · 2024-11-27
> >
> > Dear Reviewer sZQg,
> >
> > Thank you for your valuable feedback, which has greatly contributed to improving our paper. Based on your comments, we have revised Figure 2 and included a more detailed description of our model's structure as well as an experimental validation of Theorem 2.1 in the revised version of the paper.
> >
> > As the submission deadline for the revised paper is November 27, we kindly remind you to review our responses. We have made every effort to address your concerns thoroughly. We would greatly appreciate it if you could confirm whether our revisions have sufficiently resolved your questions and consider revisiting your score, or let us know if there are any further issues.
> >
> > Best regards,
> > The Authors

---

> > > ### Comment · Reviewer_sZQg · 2024-11-27
> > >
> > > thank you so much for your rebuttal. it has sufficiently addressed my concerns.

---

> ### Comment · Area_Chair_NeY9 · 2024-11-27
>
> Dear Reviewer,
>
> The authors have provided their rebuttal to your comments/questions. Given that we are not far from the end of author-reviewer discussions, it will be very helpful if you can take a look at their rebuttal and provide any further comments. Even if you do not have further comments, please also confirm that you have read the rebuttal. Thanks!
>
> Best wishes,
> AC

---

### Official Review · Reviewer_FgQz · 2024-11-04

**Soundness:** 3
**Presentation:** 3
**Contribution:** 2
**Rating:** 6
**Confidence:** 3

**Summary:**

This paper introduced MaD-Scientist, a novel approach to solving partial differential equations (PDEs) using Transformer-based models trained on Physics-Informed Neural Network (PINN) generated prior data. The key innovation is demonstrating that scientific foundation models (SFMs) can learn effectively from approximate, low-cost PINN-generated data, similar to how large language models (LLMs) learn from noisy internet data. The authors focus on convection-diffusion-reaction (CDR) equations and show impressive zero-shot inference capabilities.

**Strengths:**

- The parallel drawn between LLMs learning from noisy internet data and SFMs learning from approximate PINN-generated data is insightful and well-motivated. Similarly the zero-shot inference approach without requiring knowledge of governing equations is equally important.
- Clear mathematical formulation of the problem and the methodology.
- Extensive experiments covering multiple scenarios. The baselines are sound, and the results are excellent. Especially notable is the "superconvergence" phenomenon observed (Remark 4.1)

**Weaknesses:**

- The scope of the paper is very limited. Focus solely on 1D CDR equations may limit the broader applicability. Furthermore, no real-world applications are thoroughly explored (albeit a brief mention of semi-conductors).
- Limited discussion of computational efficiency and scaling properties, with not much variations of the transformer architecture explored.
- The appendices contain important information that should be in the main text like efficiency comparison (above point), training algorithm and some PINN failure modes.

Based on the questions/suggestions below, I am willing to increase my score.

**Questions:**

- Can the authors investigate the method's performance on higher-dimensional PDEs?
- How does the computational cost scale with problem complexity?
- Will including physical constraints in the Transformer architecture improve performance?
- A discussion of potential real-world applications would strengthen the paper

---

> ### Author Response · Authors · 2024-11-21
> **Response to reviewer FgQz**
>
> Thank you for the valuable feedback on our paper. We hope that our responses will address the reviewer’s questions effectively.
>
> **[W1, W2, Q1, Q2]** Please refer to the general response. We have added content about experiments for 2D CDR. Moreover, we have also included the computational cost analysis for 2D CDR, allowing us to observe how the computational cost increases as the dimension grows in our model.
>
>
> **[W3]** We apologize that some details are in the Appendices. The reason we prioritized including the current results in the main text was to emphasize the performance of our model in various tasks leveraging in-context learning (ICL). Due to space limitations in the main text, contents related to cost, training algorithms, and failure modes had to be moved to the Appendices.
>
>
> **[Q3]** First, thank you for your insightful question. In the extrapolation task experiment on the Burgers' equation, we apply the PINN loss as a physical constraint to our model. As a result, it demonstrates better performance compared to the original version of our model. This highlights the potential of utilizing the PINN loss as a means of improving our model’s ability to predict unexpected patterns, addressing one of the limitations of its cross-attention structure. Additional experiments supporting this approach are detailed in Appendix K. Thanks to your valuable question, we have identified a promising approach to further enhance our model in the future. Thank you again.
>
> **[Q4]** The convection-diffusion-reaction (CDR) equations studied in our work are commonly used across various physical and chemical domains. For example, they are widely utilized in environmental engineering, such as modeling pollutant dispersion in air or water and studying mixing processes in rivers [1]. They also serve as fundamental models for heat and mass transfer in fluid flows [2]. Furthermore, the reaction terms considered in our study are not arbitrary. Each reaction term is commonly used in specific domains. For instance, the Fisher reaction term describes the spatial spread of advantageous genes through populations, forming the foundation for modeling phenomena like biological invasions and genetic propagation [3]. The Allen-Cahn reaction term is a fundamental model in materials science for simulating phase separation in metals or fluids [4].
>
> Beyond the scenarios directly covered in the paper, our methodology has the potential for further expansions. For the sparse sensor data completion, our method can be applied to reconstruct missing data from sparse sensor measurements. This frequently happens for i) environmental monitoring, where filling spatial or temporal gaps in pollutant or weather data is crucial, ii) scientific simulations, where generating complete datasets from partial observations can accelerate simulations and/or improve experimental outcomes. These real-world examples and extensions demonstrate the broad applicability of our approach and provide a foundation for exploring new tasks in diverse scientific and engineering fields.
>
> [1] Jennifer G. Duan, S.K. Nanda, "Two-dimensional depth-averaged model simulation of suspended sediment concentration distribution in a groyne field", Journal of Hydrology,Volume 327, Issues 3–4 (2006).
>
> [2] POSTELNICU, Adrian. "Influence of chemical reaction on heat and mass transfer by natural convection from vertical surfaces in porous media considering Soret and Dufour effects". Heat and Mass transfer, 43.6: 595-602 (2007).
>
> [3] Fisher, Ronald Aylmer. "The wave of advance of advantageous genes." Annals of eugenics 7.4: 355-369 (1937).
>
> [4] Tourret, Damien, Hong Liu, and Javier LLorca. "Phase-field modeling of microstructure evolution: Recent applications, perspectives and challenges." Progress in Materials Science 123: 100810 (2022).

---

> > ### Author Response · Authors · 2024-11-27
> >
> > Dear Reviewer FgQz,
> >
> > Thank you for your valuable feedback, which has greatly contributed to enhancing our paper. In particular, your **[Q3]** prompted us to think about potential directions to further improve our model. Additionally, the discussion on real-world applications will serve as an important factor in strengthening our paper.
> >
> > This is a gentle reminder, as the deadline for submitting the revised paper is November 27. We have made every effort to address your questions thoroughly. We would sincerely appreciate it if you could confirm whether our responses have adequately resolved your concerns and consider revisiting your score, or let us know if you have any additional questions.
> >
> > Best regards,
> > The Authors

---

> > > ### Comment · Reviewer_FgQz · 2024-11-27
> > > **Official Response by FgQz**
> > >
> > > I would like to thank the authors for the detailed response. They have answered all my questions to satisfaction. I have increased my score to lean towards accept.

---

### Author Response · Authors · 2024-11-21
**General response (edited)**

Thank you to all the reviewers for their reviews and constructive feedback. Before addressing each review individually, we will clarify some commonly raised points in this general response. All revised parts in the revised paper have been marked in red.

1. **Experiments on 2D CDR**
We anticipated questions about experiments in higher dimensions, so we began these experiments immediately after submission. As a result, we are able to present the 2D experiment results promptly. In the revised paper, we include not only the original 1D CDR but also 2D CDR experiments. Additionally, we incorporate neural operator baselines and evaluate the model using not only the previous metrics but also the $L_{\infty}$ norm. 2D CDR experiments have been added in Appendix I of the revised paper.

2. **Experiments on Biharmonic Equation**
In response to Reviewer QAiw's question 3, we conduct additional experiments to evaluate our model's performance on PDEs with higher-order derivative terms. Specifically, we perform an interpolation task on the biharmonic equation, a PDE involving fourth-order derivative terms. Detailed information about this experiment has been added in Appendix J of the revised paper.

3. **Experiments on Burgers' Equation**
In response to Reviewer QAiw's question 10, we conduct additional experiments to evaluate our model's performance on PDE, which has a solution profile with singular behavior. Specifically, we perform an extrapolation task on the Burgers equation, a PDE involving shock formation when the viscosity parameter is small. Detailed information about this experiment has been added in Appendix K of the revised paper.

4. **Experiments on diverse reaction term with fine-tuning**
In response to Reviewer QAiw's question 14, we conduct additional experiments to fine-tune the model from Section 4.3, adapting it to each target reaction system. Detailed information about this experiment has been added in Appendix L.1 of the revised paper.

5. **Experiments on 1D CDR with diverse reaction term**
In response to Reviewer QAiw's question 11, we conduct additional experiments on 1D CDR with diverse reaction terms using fine-tuning. We evaluate our model on an interpolation task with nonzero parameters $\beta$ and $\nu$, where $\beta, \nu, \rho_1, \rho_2, \rho_3 \in [1, 5] \cap \mathbb{Z}$. We demonstrate that a model pre-trained over a broad parameter space can be effectively adapted to specific equations. Detailed information about this experiment has been added in Appendix L.2 of the revised paper.

6. **Experiments on Sensitivity to given data number**
In response to Reviewer sZQg's question 1, we conduct additional experiments on the model's sensitivity to the number of the given data $\widetilde{D}$ during testing. The results of this experiment provide empirical evidence for the findings proved in Theorem 2.1. Detailed information about this experiment has been added in Appendix M of the revised paper.

7. **Experimental details**
For all experiments, detailed hyperparameter settings have been added in Appendix N of the revised paper, and the information about computational costs has been added in Appendix C.

8. **Additional information about detail model structure**
In response to Reviewer weakness pointed out by sZQg and WwXL, we have added a detailed description of our model and a simplified figure focusing solely on the model structure in Appendix O of the revised paper. Additionally, we have incorporated a detailed illustration of the in-context learning mechanism during the inference phase into Figure 2. The refined version has been included in the revised paper.

9. **Typo**
In our original paper, we stated that we used the $L_2$​ absolute error for evaluation, but this was a typo. We actually used the $L_1$​ absolute error as our metric. We have fixed it all in the revised paper.
Additionally, Reviewer WwXL kindly pointed out a typo in Theorem 2.1 and Reviewer QAiw kindly pointed out a typo in line 372. We have corrected them in the revised paper. Furthermore, we conduct a thorough review of the manuscript to identify and correct additional typos throughout the paper to ensure overall clarity and accuracy.

---

### Author Response · Authors · 2024-11-24
**To all reviewers**

To all reviewers:

First, we would like to express our gratitude for your insightful and valuable reviews.
We have made every effort to provide the best possible responses to all the feedback we received.
Please review our responses and let us know if you have any follow-up questions.

---

### Author Response · Authors · 2024-11-27
**Additional general response**

We have prepared responses to the additional experiments that required more time and were not included in the initial rebuttal, specifically addressing General Response 5 and Reviewer QAiw's Question 11.
In particular, we conducted experiments on the 1D CDR equation with various reaction terms using fine-tuning. Our results demonstrate that a model pre-trained over a broad parameter space can be effectively adapted to specific equations. Detailed information about this experiment has been added to Appendix L.2 of the revised paper.
The rebuttal has been updated to include this information, and the related experiments have been incorporated into Section L.2 of the revised paper. As a result, the content previously in Section L has been renumbered as Section L.1.


Lastly, please note that the revised paper cannot be updated after November 27. Therefore, we kindly ask reviewers to submit any additional questions before this deadline.

---

### Meta-Review · Area_Chair_NeY9 · 2024-12-17

**Metareview:**

This paper applies LLMs to scientific foundation models (SFMs) for solving PDEs. Technically, the algorithm collects low-cost physics-informed neural network (PINN)-based approximated prior data in the form of solutions to partial differential equations (PDEs) constructed through an arbitrary linear combination of mathematical dictionaries, and utilize Transformer architectures with self and cross-attention mechanisms to predict PDE solutions without knowledge of the governing equations in a zero-shot setting. Experimental results were provided for one-dimensional convection-diffusion-reaction equations, supporting the conjecture that SFMs can improve in a manner similar to LLMs.

There were adequate discussions during the rebuttal, and the reviewers acknowledged the notable efforts from the authors during the review and AC discussion period. However, there are still general concerns:

1) The class of PDEs considered is restricted, and the results of the paper seem to be only a marginal improvement compared to previous works on SciFMs. The claim to introducing a scientific foundation model that is pre-trained on PINN-simulated data is an overclaim: while the idea of foundation models is to learn multimodel features from multiple pre-training physical systems, each involving thousands of samples with varying PDE parameters, the proposed framework is trained on only a handful of PDEs of similar types.

2) The superiority in terms of prediction accuracy over the existing scientific foundation models is not provided. Besides, the results based on the PINN prior appear to be poorer than those based on numerical priors, often showing more than 10% relative error. This raises concerns about whether the use of low-cost PINN-prior data is justified given the low prediction accuracy and can, therefore, actually outperform existing scientific foundation models.

Therefore, given the competitiveness this year at ICLR 2025, the decision is to reject this paper. Hopefully this doesn't disappoint the authors - the paper has been improving during the rebuttal, but more efforts are needed to address the aforementioned concerns for future venues. There are also many appendices newly added to the rebuttal that the authors may consider reorganizing for future versions of the paper.

**Additional Comments On Reviewer Discussion:**

As summarized above, reviewers WwXL and QAiw raised concerns on the class of PDEs considered being restricted and the superiority in terms of prediction accuracy over the existing scientific foundation models being unclear. The other two reviewers didn't further comment on this. After checking the paper in detail, these concerns are notable and result in the final decision.

---

### Decision · Program_Chairs · 2025-01-22

Reject